# PROVING OLYMPIAD INEQUALITIES BY SYNERGIZING LLMS AND SYMBOLIC REASONING

**Zenan Li**[1*], **Zhaoyu Li**[2*], **Wen Tang**[3], **Xian Zhang**[4], **Yuan Yao**[1], **Xujie Si**[2], **Fan Yang**[4], **Kaiyu Yang**[5†], **Xiaoxing Ma**[1†]

[1] State Key Lab of Novel Software Technology, Nanjing University, China,
[2] University of Toronto, [3] Peking University, [4] Microsoft Research, [5] Meta FAIR
lizn@smail.nju.edu.cn, zhaoyu@cs.toronto.edu
zhxian@microsoft.com, kaiyuy@meta.com, xxm@nju.edu.cn

---

**Problem (Chen, 2014):** If $a, b, c$ are positive reals and $a^2 + b^2 + c^2 = 1$, then

$$\frac{1}{a^2 + 2} + \frac{1}{b^2 + 2} + \frac{1}{c^2 + 2} \leq \frac{1}{6ab + c^2} + \frac{1}{6bc + a^2} + \frac{1}{6ca + b^2}. \tag{1}$$

Figure 1: We prove inequality problems in math Olympiads that involve a finite number of real variables, hypotheses, and one conclusion. Both the hypotheses and the conclusion consist of constants, variables, algebraic operations (e.g., addition, multiplication), and transcendental functions like *exp*.

## ABSTRACT

Large language models (LLMs) can prove mathematical theorems formally by generating proof steps (*a.k.a.* tactics) within a proof system. However, the space of possible tactics is vast and complex, while the available training data for formal proofs is limited, posing a significant challenge to LLM-based tactic generation. To address this, we introduce a neuro-symbolic tactic generator that synergizes the mathematical intuition learned by LLMs with domain-specific insights encoded by symbolic methods. The key aspect of this integration is identifying which parts of mathematical reasoning are best suited to LLMs and which to symbolic methods. While the high-level idea of neuro-symbolic integration is broadly applicable to various mathematical problems, in this paper, we focus specifically on Olympiad inequalities (Figure 1). We analyze how humans solve these problems and distill the techniques into two types of tactics: (1) scaling, handled by symbolic methods, and (2) rewriting, handled by LLMs. In addition, we combine symbolic tools with LLMs to prune and rank the proof goals for efficient proof search. We evaluate our framework on 161 challenging inequalities from multiple mathematics competitions, achieving state-of-the-art performance and significantly outperforming existing LLM and symbolic approaches without requiring additional training data.

## 1 INTRODUCTION

Automated theorem proving has been a long-standing goal in AI (Newell & Simon, 1956). Recent research explores leveraging large language models (LLMs) to generate formal proofs that can be verified in formal proof systems like Lean (de Moura et al., 2015), opening a new avenue to theorem proving (Li et al., 2024; Yang et al., 2024). This promising approach has already led to tools that assist human mathematicians (Song et al., 2024) and the first AI that achieves silver-medal performance in the International Mathematical Olympiad (IMO) (Google DeepMind, 2024).

While LLM-based proof generation shows great promise across various mathematical domains, its performance is constrained by the scarcity of formal proof data. Furthermore, it remains an open

---

[*]Equal contribution. This work was partially done during Zenan's internship at MSRA.
[†]Equal advising. All experiments were conducted outside Meta's compute infrastructure.

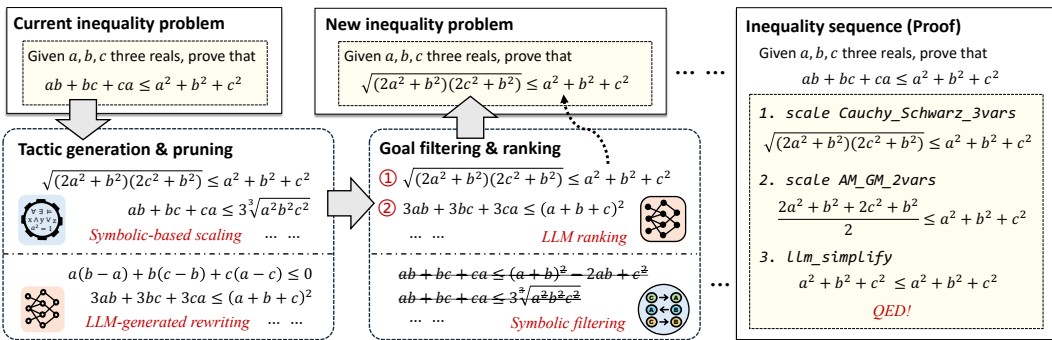

Figure 2: An overview of our neuro-symbolic inequality prover LIPS. By integrating both LLMs and symbolic methods in an iterative process of tactic generation and goal selection, it can generate human-readable and formally verifiable proofs in Lean for Olympiad-level inequality problems.

problem whether LLMs can perform precise and complex symbolic manipulations (Hammond & Leake, 2023). To address these limitations, mechanical symbolic reasoning is still essential. Unlike LLMs, symbolic methods leverage domain-specific knowledge to achieve greater efficiency and generalization without relying on extensive training data (Wu, 2008; Heule et al., 2016). Integrating LLMs with symbolic methods presents a promising strategy for tactic generation and theorem proving. This raises a key question: *Which aspects of mathematical reasoning are best suited to LLMs, and which to symbolic methods?* By exploring this question, we aim to combine the strengths of both approaches effectively. Since symbolic methods are inherently domain-specific, we focus on a concrete domain: inequalities, which offers a balance between feasibility and practicality.

**Mathematical Inequalities.** Inequalities arise from various branches of mathematics (Hardy et al., 1952). Their proofs often rely on a relatively small set of fundamental techniques, yet these techniques can be applied in remarkably intricate and nuanced ways. In this paper, we further constrain our scope to elementary algebraic inequalities (Figure 1), which were prevalent in high-school mathematics competitions (Manfrino et al., 2010), e.g., Problem 2 in IMO 2000. Additionally, these inequalities are closely related to quantifier-free real arithmetic in satisfiability modulo theories (SMT) (Barrett & Tinelli, 2018), which have numerous practical applications in formal verification.

Even this restricted class of inequalities poses significant challenges that surpass the capabilities of current symbolic or neural provers. Symbolic methods (Yang, 1999; Uray, 2020) partition the variable space (e.g., $\mathbb{R}^3$ for three variables) into a finite number of cells, which are then exhaustively enumerated. These approaches suffer from combinatorial explosion and quickly become computationally infeasible for competition-level problems. Furthermore, due to their enumerative nature, these methods are black boxes that cannot produce human-readable proofs. On the other hand, neural approaches (Wu et al., 2021; Wei et al., 2024) fine-tune language models to generate formal proofs by predicting tactics or evaluating subgoals during proof search. However, due to data scarcity and difficulties in generalization, they frequently underperform on certain types of problems.

**Our Approach.** We introduce LIPS (LLM-based inequality prover with symbolic reasoning), a neuro-symbolic framework that synergistically combines LLMs with domain-specific symbolic techniques, as illustrated in Figure 2. Specifically, we analyze common proving strategies used by humans in inequality proofs and categorize them into two types of tactics: *scaling* and *rewriting*. Scaling tactics apply existing lemmas (e.g., the Cauchy-Schwarz inequality) to scale a subterm in the current goal. The set of lemmas is finite, and each lemma can be applied only in a limited number of ways. Therefore, we can enumerate all possible scaling tactics using symbolic tools. However, not all scaling tactics are useful for the current goal; over-scaling may render the goal invalid. We filter out such invalid scaling by using symbolic tools such as SMT solvers to check for counterexamples of the resulting goals. Rewriting tactics, on the other hand, transform a term into an equivalent form (e.g., subtracting $2ab$ from both sides of the current goal). Any term can be rewritten in infinite ways, making exhaustive enumeration impossible. To address this, we use LLMs to generate rewriting tactics by designing a series of prompts for different rewriting formats. By leveraging the mathematical intuition embedded in LLMs, we implicitly prune the infinite tactic space, sampling the most promising equivalent transformations.

Scaling and rewriting the current goal leads to a set of subgoals with new inequalities to prove. Efficient proof search requires prioritizing the most promising subgoals for further exploration. To this end, we employ two strategies: *symbolic filtering* and *neural ranking*. In symbolic filtering, we use heuristics based on inequalities' homogeneity and decoupling properties to filter out unpromising subgoals. In neural ranking, the remaining subgoals are fed into an LLM, which compares and ranks them using chain-of-thought prompting (Wei et al., 2022). After filtering and ranking, we end up with a small number of ranked subgoals. We then iteratively generate and apply new tactics, filter and rank the resulting subgoals, until the final goal becomes trivially provable, resulting in a proof that is both human-readable and formally verified by Lean.

We evaluate LIPS on 161 challenging inequalities collected from three problem sets (Chen, 2014; Tung, 2012; Wei et al., 2024). The experimental results show that LIPS consistently achieves state-of-the-art performance and significantly outperforms existing neural and symbolic methods in both effectiveness and speed. Remarkably, out of 61 inequality problems sourced from various math Olympiad competitions, LIPS successfully proves 56 within 90 minutes, whereas the previous best approach proves only 41 problems and fails to generate human-readable proofs.

## 2 INEQUALITY THEOREM PROVING

In this section, we briefly review existing neural and symbolic methods for inequality proving. In a nutshell, symbolic methods complete proofs by exploring the problem space exhaustively, whereas neural methods use LLMs to sample proof steps from a vast tactic space. Each approach comes with distinct advantages and challenges.

### 2.1 SYMBOLIC METHODS

Consider a set of polynomial inequalities $\Phi_j \bowtie 0, j = 1, \ldots, m$, where $\bowtie \in \{\leq, <, \geq, >, =, \neq\}$, and each polynomial is defined over $n$ variables as $\Phi_j(x_1, \ldots, x_n) := \sum_{i=1}^{k} a_i \cdot x_1^{i_1} x_2^{i_2} \cdots x_n^{i_n}$, cylindrical algebraic decomposition (CAD) (Arnon et al., 1984; Caviness & Johnson, 2012; Kremer, 2020) divides the variable space $\mathbb{R}^n$ into multiple connected semi-algebraic sets, known as *cells*. Within each cell, the sign of every polynomial remains constant (positive, negative, or zero). Figure 6(a) illustrates this process with an example of CAD-producing cells for two intersecting unit circles. By exploiting this sign invariance, we can determine the satisfiability of inequalities by enumerating and checking all the cells, rather than searching the entire infinite space $\mathbb{R}^n$.

CAD and its variants have been extensively utilized in modern SMT solvers (Jovanović & de Moura, 2013; Kremer et al., 2022; Uncu et al., 2023). For inequality theorem proving, CAD transforms the problem into an enumeration task, exhaustively examining all cells to assess whether the inequality can be satisfied. While it is capable, CAD's performance remains unsatisfactory for several reasons. Firstly, CAD's heuristic strategies are primarily designed to efficiently find counterexamples rather than to optimize the proof search process. Secondly, CAD's proving mechanism fails to generate explicit and interpretable reasoning paths, hindering both automatic verification by existing interactive theorem provers and human interpretation of the proofs. Additionally, CAD suffers from double exponential computational complexity relative to the number of variables $n$ (Davenport & Heintz, 1988), causing its efficiency to decrease significantly as the number of variables increases. Specifically, when dealing with nonlinear inequalities involving fractions or radical expressions, auxiliary variables are always introduced to eliminate these terms (e.g., converting $\sqrt{x_1} + \sqrt{x_2} + x_3^2 = 0$ into $\{x_4 + x_5 + x_3^2 = 0, x_4^2 = x_1, x_5^2 = x_2, x_4 > 0, x_5 > 0\}$). Although this transformation successfully rewrites the inequality into polynomial form, it drastically degrades the performance of CAD.

### 2.2 NEURAL METHODS

In contrast to symbolic methods based on CAD, some approaches leverage neural networks to predict tactics within an interactive theorem prover, generating human-like, step-by-step formal proofs. Specifically, these inequality proofs are typically structured in a top-down sequential manner, where each tactic either transforms the current goal into a new subgoal or directly completes the proof (see Figure 6(b)). Figure 6(c) provides an example proof of the inequality $ab + bc + ca \leq a^2 + b^2 + c^2$.

Among existing work, INT (Wu et al., 2021) designs a theorem generator for elementary-level inequalities by randomly sampling axioms from a fixed set. It trains a Transformer (Vaswani et al.,

2017) model to predict tactics and utilizes another value network with the Monte Carlo tree search to complete proofs. Similarly, AIPS (Wei et al., 2024) implements a synthetic generator that can produce IMO-level inequalities and trains a language model to score each inequality expression in a curriculum manner, performing a best-first search to solve these problems. However, their inequality generators have some restrictions – either limited in difficulty or constrained by specific forms like cyclic symmetry. Moreover, these approaches mainly rely on fine-tuning models on large-scale datasets, making them highly dependent on the quantity and diversity of the training data.

## 3 TACTIC GENERATION AND PRUNING

Compared to general theorem proving, the tactics used in inequality proofs are more well-structured: Common tactics can be categorized into two types, namely *scaling* and *rewriting* (Lee, 2004; Ikenaga, 2018). Scaling refines the given inequality using a known inequality lemma, such as the arithmetic and geometric means (AM-GM) inequality, while rewriting transforms a given inequality into an equivalent form, such as multiplying both sides by two. Scaling and rewriting have distinct characteristics. The number of lemmas that can be used in scaling is finite, whereas the number of equivalent transformations in rewriting can be infinite, which leads to different strategies for tactic generation. Furthermore, unlike rewriting, scaling does not preserve the provability of the inequality, leading to different strategies for pruning the resulting subgoals.

Let's consider the inequality problem in Figure 1 as a running example. To the best of our knowledge, the proof for this problem is not readily available online, and neither the most advanced LLMs (e.g., OpenAI o3-mini and DeepSeek-R1) nor existing CAD solvers can solve this inequality.

### 3.1 SCALING TACTICS

Given an inequality lemma (e.g., the special case of the AM-GM inequality $u^2 + v^2 \geq 2uv$), we enumerate all possible ways to instantiate its arguments ($u$ and $v$) so that it matches a part of the current proof goal. This is a classical problem called e-matching (De Moura & Bjørner, 2007; Moskal et al., 2008) and can be solved using existing symbolic tools (Willsey et al., 2021). However, it introduces a challenge due to the potentially large number of possible patterns. In our running example, there are a total of 162 possible pattern matches for the two-variable AM-GM inequality, including cases like $\{u := a, v := \sqrt{2}\}$, $\{u := 1, v := \frac{1}{\sqrt{a^2+2}}\}$.

Since scaling tactics refine the inequality goal, they may produce potentially incorrect subgoals (i.e., unprovable statements). For instance, applying the AM-GM inequality with the pattern $\{u := a, v := \sqrt{2}\}$ would transform the original inequality (1) into

$$\frac{1}{2\sqrt{2}a} + \frac{1}{2\sqrt{2}b} + \frac{1}{2\sqrt{2}c} \leq \frac{1}{6ab + c^2} + \frac{1}{6bc + a^2} + \frac{1}{6ca + b^2}. \tag{2}$$

However, this inequality does not hold when $a = b = c = \frac{\sqrt{3}}{3}$. Specifically, out of the 162 possible patterns for applying the AM-GM inequality in our running example, only six (i.e., $\{u := a, v := b\}$ and its symmetric or cyclical versions) yield correct deductions. Therefore, we propose using CAD to identify counterexamples in the new inequality goals and eliminate the scaling tactics that produce them. Moreover, since most scaling tactics are incorrect and thus induce counterexamples, CAD can efficiently detect and discard them using well-established heuristic strategies.

To further enhance the efficacy and efficiency of scaling tactic pruning, we propose several additional methods to complement CAD in searching for counterexamples:

**Quick Check via Test Cases.** When CAD identifies a counterexample for a scaling tactic, we store it as a "test case". For any subsequent scaling tactic, we will perform a quick check using this test case before invoking CAD to determine if the counterexample invalidates the tactic.

**Incorporating Numerical Optimization.** Since most inequality problems are differentiable, we also use gradient-based optimization as an effective alternative when CAD fails. Specifically, we rewrite the inequality $f(x) \leq g(x)$ into $\min_x[g(x) - f(x)]$, and then integrate Newton's method with simulated annealing to solve it (Fu & Su, 2016; Ma et al., 2019; Ni et al., 2023).

**Utilization of Prior Knowledge.** Additionally, prior knowledge can be leveraged in scaling tactic pruning when the specific form of an inequality problem is known. For example, the inequalities generated and evaluated by AIPS (Wei et al., 2024) are cyclically symmetric and sufficiently tight to achieve equality. Consequently, AIPS verifies the consistency of equality conditions across multiple tactics and discards subgoals that either violate these conditions or do not conform to the desired form. However, to accommodate a broader range of inequalities, our current framework does not incorporate such priors, even though this information could efficiently prune scaling tactics.

## 3.2 REWRITING TACTICS

After pruning, only a few scaling tactics are applicable. For example, we may choose to apply the two-variable AM-GM inequality (with pattern $\{u := a, v := b\}$ and its cyclical versions) and derive a new proof goal from the initial goal (1), formulated as

$$\frac{1}{a^2+2} + \frac{1}{b^2+2} + \frac{1}{c^2+2} \leq \frac{1}{3a^2+3b^2+c^2} + \frac{1}{3b^2+3c^2+a^2} + \frac{1}{3a^2+3c^2+b^2}. \quad (3)$$

To solve the current proof goal, we need some equivalent transformations. For example, one may simplify the proof goal by replacing $\frac{1}{3a^2+3b^2+c^2}$ with $\frac{1}{3-2c^2}$ using the assumption $a^2 + b^2 + c^2 = 1$. However, figuring out effective ways to transform the goal needs creativity and cannot be readily automated through brute-force methods. Consequently, relying solely on symbolic pattern matching tends to be highly ineffective. Moreover, the argument space for rewriting tactics is typically infinite. For example, the assumption $a^2 + b^2 + c^2 = 1$ can be inserted at almost any point within the inequality, significantly enlarging the space of possible rewriting tactics.

To generate and prune the rewriting tactics, we propose directly prompting an LLM to generate candidates. Specifically, for different types of rewriting (e.g., rearrangement, simplification, denominator cancellation, etc.), we design tailored prompts that guide the LLM to transform the current proof goal into an appropriate form. The details of these prompts can be found in Appendix C. In this setting, tactic pruning is performed implicitly leveraging the algebraic intuition of LLMs, thereby effectively reducing the space of possible rewriting tactics.

Besides neural-guided rewriting, we incorporate two additional symbolic rewriting tactics: the sum-of-squares (Chen, 2013) and the tangent line (Li, 2005) tricks. The sum-of-squares trick attempts to transform the current expression into a summation of non-negative terms, e.g., proving $2x^2 + 2xy - 3x + y^2 + \frac{9}{4} \geq 0$ by recognizing $2x^2 + 2xy - 3x + y^2 + \frac{9}{4} = (x + y)^2 + (x - \frac{3}{2})^2$. The tangent line trick is a powerful variable substitution technique that is frequently used to tackle inequality problems in math competitions.

## 4 GOAL FILTERING AND SELECTION

Scaling and rewriting tactics yield a collection of new proof goals. By combining these newly generated goals with existing unexplored ones, we can derive a candidate set. The next step is to select the most promising goal from this candidate set for subsequent proving. In our running example, we can derive 16 new proof goals from the current goal (3). Below, we present three of these new goals, each derived by applying the two-variable AM-GM inequality, the three-variable Titu inequality, and simplifying using the given assumption $a^2 + b^2 + c^2 = 1$, respectively:

$$\frac{1}{a^2+2} + \frac{1}{b^2+2} + \frac{1}{c^2+2} \leq 2\sqrt{\frac{1}{a^2+3b^2+3c^2} \cdot \frac{1}{b^2+3a^2+3c^2}} + \frac{1}{c^2+3a^2+3b^2}; \quad (4)$$

$$\frac{1}{a^2+2} + \frac{1}{b^2+2} + \frac{1}{c^2+2} \leq \frac{9}{7a^2+7b^2+7c^2}; \quad (5)$$

$$\frac{1}{a^2+2} + \frac{1}{b^2+2} + \frac{1}{c^2+2} \leq \frac{1}{3-2a^2} + \frac{1}{3-2b^2} + \frac{1}{3-2c^2}. \quad (6)$$

To select proof goals effectively, existing methods often train a language model as a value function to evaluate and rank each goal in the candidate set. However, limitations in data quantity and diversity can significantly degrade the performance of these fine-tuned models. For example, the largest

collected inequality dataset in Lean comprises only 46K samples (Ying et al., 2024), while the largest synthesized dataset of competition-level inequalities contains merely 191K cyclically symmetric instances (Wei et al., 2024). Consequently, we opt to directly employ an off-the-shelf LLM (e.g., GPT-4o) without fine-tuning. However, as more proof goals accumulate, the context length required for prompting an LLM for goal ranking becomes substantial, potentially leading to the issue of LLMs being "lost in the middle" (Liu et al., 2024). To address this, we propose dividing the goal selection pipeline into two stages: symbolic filtering and neural ranking.

## 4.1 SYMBOLIC FILTERING

First, we eliminate proof goals that are less promising based on carefully designed symbolic rules. Specifically, we prioritize two key properties: (1) *homogeneity*, meaning both sides of the inequality have the same degree (e.g., $a^2 + b^2 \geq 2ab$); and (2) *decoupling*, which refers to whether the inequality contains mixed-variable terms (e.g., $abc$ is considered coupled). Both properties are reasonable criteria for prioritization. Regarding homogeneity, most substitution and transformation tactics preserve homogeneity. Hence, a homogeneous inequality allows for a broader range of tactics to be applied, thereby producing more valuable proof goals. For decoupling, an inequality with fewer coupled terms is not only clearer but also amenable to a greater variety of techniques, such as the sum-of-squares and tangent line tricks.

To measure decoupling and homogeneity, we first approximate the proof goal by a polynomial inequality using Taylor expansion. We then compute the expectation of the number of variables in each term and the variance of the degree of each term, respectively. Formally, given a polynomial inequality expressed as $\Phi(x_1, \ldots, x_n) = \sum_{i=1}^{k} a_i \cdot x_1^{i_1} x_2^{i_2} \cdots x_n^{i_n} \leq 0$, the decoupling score (DC) and homogeneity score (HM) are computed as:

$$\text{DC}(\Phi) = \frac{1}{k} \sum_{i=1}^{k} a_i \left( \sum_{j=1}^{n} \mathbb{I}(i_j > 0) \right), \quad \text{HM}(\Phi) = \frac{1}{k} \sum_{i=1}^{k} \left( d_i - \frac{1}{k} \sum_{j=1}^{k} d_j \right)^2,$$

where $d_i = i_1 + \cdots + i_n$ is the total degree of $i$-th term in the inequality.

It is worth noting that the homogeneity score and the decoupling score are not always consistent. In our running example, the newly generated goals in (4), (5), and (6) achieve homogeneity scores of 0.56, 0.55, and 0.80, and decoupling scores of 0.44, 0.48, and 0.66, respectively. As a result, we normalize the scores into $[0, 1]$ and then compute the average score to filter the candidates.

## 4.2 NEURAL RANKING

Symbolic rules are not universally effective. Hence, we use these rules solely to eliminate unpromising proof goals, leaving top-$k$ candidates, for final selection by an LLM. Unlike symbolic filtering, which requires explicit definitions of inequality metrics, we use the chain-of-thought prompting (Wei et al., 2022; Chu et al., 2023) to query an LLM to rank the proof goals based on their proving difficulty. The detailed prompt for the running example is shown below.

---

**Prompt of neural ranking**

I am trying to prove the original inequality: " If $a, b, c$ are positive reals and $a^2 + b^2 + c^2 = 1$, then $\frac{1}{a^2+2} + \frac{1}{b^2+2} + \frac{1}{c^2+2} \leq \frac{1}{3a^2+3b^2+c^2} + \frac{1}{3b^2+3c^2+a^2} + \frac{1}{3a^2+3c^2+b^2}$", and transform it into the following inequalities.
(1) $\frac{1}{a^2+2} + \frac{1}{b^2+2} + \frac{1}{c^2+2} \leq 2\sqrt{\frac{1}{a^2+3b^2+3c^2} \cdot \frac{1}{b^2+3a^2+3c^2}} + \frac{1}{c^2+3a^2+3b^2}$
(2) $\frac{1}{a^2+2} + \frac{1}{b^2+2} + \frac{1}{c^2+2} \leq \frac{9}{7a^2+7b^2+7c^2}$
(3) $\frac{1}{a^2+2} + \frac{1}{b^2+2} + \frac{1}{c^2+2} \leq \frac{1}{3-2a^2} + \frac{1}{3-2b^2} + \frac{1}{3-2c^2}$
Your task is to rank the transformation results in a descent order. Note that
1. Please reason step by step;
2. More meaningful transformation, i.e., reduce the proving difficulty, should be ranked higher;
3. Put the index of selected inequality within \\boxed{{}}, e.g., \\boxed{{(1),(2),(3)}}.

---

Table 1: Proof success rates (%) on three datasets. LIPS consistently outperforms baselines.

| Dataset | # of Problems | Neural Provers | | | Symbolic Provers | | LIPS | Δ |
|---------|---------------|------|-------|--------------------|------------------|------------------|------|------|
| | | DSP | MCTS | AIPS[†] | CAD[‡] | MMA[‡] | | |
| ChenNEQ | 41 | 0.0 | 17.0 | - | 70.7 | 68.2 | **95.1** | 24.4↑ |
| MO-INT | 20 | 0.0 | 15.0 | 50.0 | 60.0 | 60.0 | **80.0** | 20.0↑ |
| 567NEQ | 100 | 0.0 | 4.0 | - | 54.0 | 52.0 | **68.0** | 14.0↑ |
| Total | 161 | 0.0 | 8.6 | - | 59.0 | 57.1 | **76.3** | 17.3↑ |

[†] We include the results reported by the AIPS paper since the code is not publicly available.
[‡] CAD and MMA only output verification results, they cannot produce human-readable proofs.

## 5 EXPERIMENTS

In this section, we conduct a series of experiments to address the following three research questions:

**RQ1: Efficacy** – Can LIPS prove more problems compared to existing methods?

**RQ2: Efficiency** – Can LIPS obtain proofs in less time compared to existing methods?

**RQ3: Scalability** – Can LIPS be improved by using more scaling lemmas or more powerful LLMs?

### 5.1 EXPERIMENTAL SETUP

**Datasets.** We evaluate LIPS on three datasets: ChenNEQ, MO-INT-20, and 567NEQ, respectively. ChenNEQ consists of 41 Olympiad-level inequalities collected by Chen (2014); MO-INT is a new competition-level inequality benchmark introduced in AIPS (Wei et al., 2024), featuring 20 problems sourced from IMO shortlists and various national mathematical Olympiads; 567NEQ consists of 567 hard inequalities created by Tung (2012) and we randomly selected 100 problems from the original problem set as the testbed for our framework. To formalize the problems in Lean 4, we directly translate the LaTeX source code into Lean using manually defined rules.

**Baselines.** We compare LIPS with five baselines: DSP (Jiang et al., 2023b), CAD (Kremer, 2020), MMA (Wolfram Research), MCTS (Wu et al., 2021), and AIPS (Wei et al., 2024). DSP consists of two steps, natural language reasoning generation and proof autoformalization, and we instantiate the LLM used in each step GPT-4o. MCTS (Monte Carlo tree search) has been explored in previous studies (Wu et al., 2021) and serves as an alternative method for proof goal selection. AIPS is an inequality prover system based on SymPy, which has demonstrated the capability to prove competition-level inequalities. CAD integrates a series of CAD-based inequality solvers including Z3 (De Moura & Bjørner, 2008), CVC5 (Kremer et al., 2022), RC-CAD (Lemaire et al., 2005), and Bottema (Lu, 1998). MMA, referring to Mathematica, incorporates the CAD algorithm with other reduction strategies, providing a powerful algebraic system for inequality verification (Wolfram Research, 2020). Further implementation details for the baseline methods are provided in Appendix C.

**Implementation.** The detailed processes of tactic generation, goal selection, as well as the overall framework are provided in Appendix B. To construct the tactics, we design a total of 96 scaling tactics and 16 rewriting tactics, each formalized in Lean 4. The corresponding premises and LLM prompts are summarized in Appendix C. For the counterexample search in scaling tactic pruning, we integrate four CAD-based solvers (Z3, CVC5, RC-CAD, and Bottema) and implement an optimizer based on SciPy (Virtanen et al., 2020). For the LLM involved in transformation tactic generation and proof goal ranking, we use GPT-4o (version Azure-0501). In symbolic filtering, we fix the size of the filtered goal set to 10, as it is the largest size that ensures GPT-4o's efficacy. The code, together with the experimental data, is available at `https://github.com/Lizn-zn/NeqLIPS`.

### 5.2 EXPERIMENTS

**RQ1 : Efficacy.** We evaluate the proof success rates of LIPS and the five comparative methods across the three datasets. For each proving task, a time limit of 90 minutes is imposed, consistent with that of AIPS and the standard problem-solving time constraint in the IMO. The overall results are presented in Table 1. First, we observe that the neural methods (DSP and MCTS) cannot achieve satisfactory performance. An analysis of DSP's results reveals that GPT-4o is unable to provide ac-

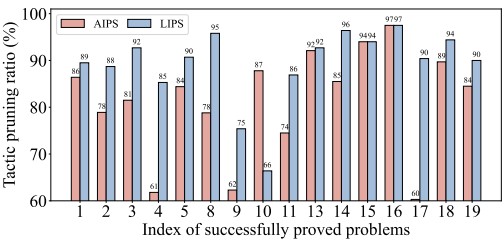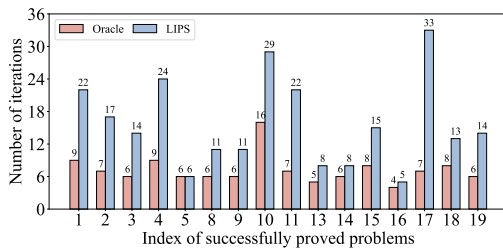

Figure 3: Tactic pruning ratio (↑, higher is better) and number of iterations (↓). The results illustrate that the tactic pruning method of LIPS is very stable, and outperforms the existing method by 7.92% on average. Furthermore, the high efficiency of LIPS is derived from accurate goal selection, allowing a proof to be successfully constructed with a small number of goal selection iterations.

curate natural language solutions or generate precise formal proofs in the Lean 4 language, resulting in a zero success rate. Alternatively, MCTS struggles to effectively identify the correct reasoning path among numerous proof goals, causing many proving attempts to terminate due to timeouts. We also evaluate the performance of the recent OpenAI o1-preview model on the MO-INT dataset. Through manual inspection of the generated natural language answers, we find that none of the problems are correctly solved. Examples of these neural methods are provided in Appendix E.

Symbolic provers outperform neural provers, CAD and MMA achieve overall success rates of 59.0% and 57.1%, respectively. However, LIPS further surpasses symbolic provers by a significant margin of 14.0% to 24.4%. Notably, symbolic provers fail to produce any human-readable reasoning path. In contrast, the proofs generated by LIPS are not only accessible and human-readable, but also have been successfully verified by the Lean theorem prover. For reference, we include two examples (one showcasing a successful proof and the other demonstrating a failed attempt) in Appendix E.

**RQ2 : Efficiency.** Since existing methods are built on different deduction engines, a direct comparison of their proving time could be unfair. Instead, we break down the efficiency evaluation into two aspects, i.e., the pruning ratio of scaling tactics and the number of iterations in goal selection. Given that AIPS uses the equality check as the scaling tactic pruning strategy, we compare this approach with the CAD-based strategy employed in LIPS. Figure 3 provides the result of each problem in the MO-INT dataset. LIPS outperforms the existing method in 13 out of 20 problems, and achieves an average improvement of 7.92%.

Furthermore, we count the number of goal selection iterations for each successfully proved problem, and present the results in Figure 3. Due to the absence of a comparison method, we only include the oracle (i.e., optimal goal selection) as a reference. We can observe that LIPS performs no more than 33 search loops to successfully obtain a proof, and for 12 out of 16 problems, it exceeds the oracle by fewer than 10 steps. In addition, LIPS requires an average of 15.75 search loops, which is only 2.17 times that of the oracle (7.25), demonstrating that LIPS's efficiency also stems from its high accuracy in generating proving paths.

**RQ3 : Scability.** We conduct four experiments on the ChenNEQ dataset to explore the scalability of LIPS's symbolic and neural components. For the symbolic part, we first examine how expanding the scaling tactics affects the performance of LIPS. To this end, we randomly select 7 sets of scaling tactics with varying sizes and plot the performance curve in Figure 4(a). The results show that the proof success rate consistently increases as more scaling tactics are included, suggesting potential benefits in further enlarging our scaling tactic library. The second experiment investigates the effect of different sizes of the filtered set. The corresponding performance curve is shown in Figure 4(b). We observe that the success rate remains robust (over 85%) with sizes from 8 to 16, but decreases significantly when the set is either too small or too large.

For the neural part, we explore the performance of LIPS with different LLMs serving in rewriting tactic generation and neural ranking. We select three alternative LLMs, i.e., Mathstral 7B (Jiang et al., 2023a), LLaMA-3 8B (AI, 2024), and DeepSeek-chat V2.5 (DeepSeek-AI, 2024). We also include a baseline method as an ablative study. In rewriting tactic generation, the baseline uses SymPy `simplify` function instead of existing LLM-based rewriting tactics. In neural ranking, we directly use random selection as the baseline. The proving success rates are provided in Figure 5. The re-

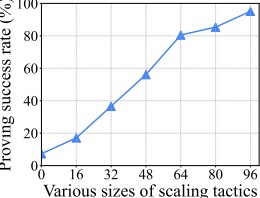 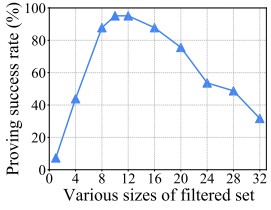 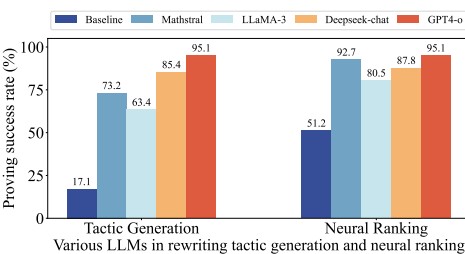

Figure 4: Performance curves across different scaling tactics and filtered set sizes. Results indicate that: (1) Adding scaling tactics consistently improves the proving success rate; (2) The effectiveness of symbolic filtering and the size of the filtered set are critical factors.

Figure 5: Performance of various LLMs as alternatives. Results illustrate that: (1) LLMs are crucial in rewriting tactic generation and neural ranking; (2) The more powerful LLM can achieve higher proving success rate.

sults demonstrate that all three alternative LLMs exhibit strong mathematical intuition, achieving performance comparable to GPT-4o in neural ranking. However, there exists a small decline in performance for rewriting tactic generation, which may be due to differences in instruction following and mathematical reasoning capabilities.

**Ablation study.** To showcase the strength of our neuro-symbolic paradigm, we present the results of removing neural or symbolic modules of LIPS in Appendix D. We also analyze the performance of symbolic solvers, offering guidance on optimal time limit settings in LIPS.

**Case study.** Besides the running example (1), we provide two additional examples in Appendix F to illustrate that LIPS can discover new proofs, which are previously unavailable online. Moreover, we also present two examples in Appendix F to demonstrate that users can verify human-written proofs by comparing them with the reasoning paths generated by LIPS.

# 6 RELATED WORK

**Symbolic Tools for Mathematical Reasoning.** Symbolic tools are essential for performing exact computations and formal reasoning in mathematics. Interactive theorem provers such as Isabelle (Paulson, 1994), Coq (Coq, 1996), and Lean (de Moura et al., 2015) enable users to build verifiable proofs manually, ensuring correctness through rigorous formal logic. These systems have been instrumental in formalizing and verifying significant mathematical theorems, including the Four Color Theorem (Gonthier, 2008) and the Kepler Conjecture (Hales et al., 2017). Alternatively, some symbolic reasoning tools aim to solve mathematical problems without human intervention. Automated theorem provers like E (Schulz, 2002) and Vampire (Kovács & Voronkov, 2013) are designed to prove mathematical statements by systematically exploring possible proofs within a logical framework, particularly excelling in first-order logic. SMT solvers such as Z3 (De Moura & Bjørner, 2008) and CVC5 (Barbosa et al., 2022) determine the satisfiability of logical formulas with respect to background theories like arithmetic, bit-vectors, and arrays by integrating logical reasoning with theory-specific decision procedures. Computer algebra systems like Mathematica (Wolfram Research), Maple (Heck & Koepf, 1993), and SymPy (Meurer et al., 2017) manipulate mathematical expressions symbolically, supporting functionalities such as simplification, differentiation, integration, and equation solving. Despite their capabilities, these automated solvers struggle with competitive mathematical problems and often cannot generate human-readable reasoning steps.

**Machine Learning for Formal Theorem Proving.** There is a longstanding tradition of leveraging machine learning techniques to automate theorem proving (Urban et al., 2008; Gauthier et al., 2017; Zhang et al., 2021; Piotrowski et al., 2023; Blaauwbroek et al., 2024). Recently, the emergence of LLMs has expanded the potential of these techniques, offering new opportunities for automating theorem proving (Li et al., 2024; Yang et al., 2024). A line of research (Polu & Sutskever, 2020; Wu et al., 2021; Han et al., 2022; First et al., 2023; Yang et al., 2023; Xin et al., 2024a; Lin et al., 2025) fine-tunes pre-trained language models on large-scale formal datasets to predict the tactics given a proof goal. Alternative approaches (Jiang et al., 2023b; Xin et al., 2024b; Zhao et al., 2024; Zheng et al., 2024; Thakur et al., 2024) integrate LLMs into structured prompt frameworks for formal theorem proving, leveraging information such as natural language proofs or feedback from

interactive theorem provers. Some methods (Polu & Sutskever, 2020; Lample et al., 2022; Polu et al., 2023; Wang et al., 2023; Wei et al., 2024) also train LLMs as value networks to evaluate each subgoal and guide the proof search to completion. However, these methods primarily rely on LLMs to prune the search space, and their performance heavily depends on the quality and diversity of the training data, which may limit their generalizability to novel or more complex mathematical problems. A notable exception and closely related work to ours is AlphaGeometry (Trinh et al., 2024), which uses a language model only to predict auxiliary construction rules, after which its symbolic solver automatically enumerates all inference rules to generate the proof in a specialized language used in GEX (Chou et al., 2000). Compared to their approach, our neuro-symbolic framework generates step-by-step proofs in general-purpose formal language Lean and allows for integrating arbitrary symbolic tools that do not require direct proof generation. Furthermore, our focus on inequality problems spans a much wider range of complex mathematical skills than plane geometry problems. Our techniques could serve as a solid foundation for broader areas of mathematical research such as information theory (Dembo et al., 1991), optimization (Nesterov, 2013), and deep learning (Roberts et al., 2022), making it a suitable pathway to more advanced mathematical problems.

## 7    LIMITATIONS AND FUTURE WORK

While LIPS has shown significant promise in generating formal proofs for Olympiad inequalities, several avenues remain open for enhancement and expansion.

**Automating the Formalization of Tactics.**   Our framework currently relies on a set of manually crafted tactics for scaling and rewriting inequalities, such as various forms of AM-GM inequality. This manual effort may impact scalability, given that the effectiveness of our approach is closely tied to the breadth of available tactics. Future work could focus on automating the discovery, formalization, and proof of new tactics to expand the tactic library. Developing methods for automatic tactic generation would reduce human effort and enhance the framework's scalability and adaptability.

**Enhancing the Reasoning Capabilities of LLMs.**  We leverage the mathematical insights learned by LLMs in our framework, and there is potential to further improve their reasoning performance. One promising direction is to collect or generate additional formal inequality problems and their corresponding proofs to create a richer dataset for fine-tuning LLMs specifically for this task. Some existing techniques (Li et al., 2025) may be useful for generating diverse and high-quality problems to enhance the LLMs' capabilities in handling inequalities, leading to better overall performance.

**Broadening the Application Domain.**  While our framework currently focuses on Olympiad-level elementary algebraic inequalities, extending it to more complex problems, such as concentration inequalities in machine learning theory, presents an exciting avenue for future research. This would involve improving the symbolic solver to handle inequality structures that consist of infinite variables and higher-order concepts like expectations or variances. Developing efficient algorithms and symbolic reasoning methods for these advanced mathematical constructs could significantly broaden the applicability of our neuro-symbolic paradigm. Extending our approach to other mathematical domains holds great potential and is a promising direction for future work.

## 8    CONCLUSION

In this paper, we introduce a neuro-symbolic framework for generating formal proofs that integrates the mathematical intuition learned by LLMs with domain-specific insights encoded by symbolic methods, specifically focusing on the domain of Olympiad inequalities. We categorize the tactics used in inequality proofs into two types: scaling and rewriting. Symbolic methods are employed to generate and filter scaling tactics by applying a set of lemmas through mechanical symbolic reasoning. LLMs are leveraged to generate rewriting tactics, implicitly pruning the infinite number of equivalent transformations to a manageable set. We further combine symbolic tools with LLMs to prune and rank subgoals, enhancing the efficiency of proof search. Experiments on challenging inequalities from three problem sets show that our neuro-symbolic inequality prover LIPS significantly outperforms both LLMs and symbolic methods, demonstrating the effectiveness of the neuro-symbolic integration and laying a solid foundation for its adoption in broader domains.

ACKNOWLEDGMENT

We appreciate anonymous reviewers for their valuable comments and engaging discussions. This work was partially conducted during Zenan's and Wen's internships at MSRA. It is supported by the National Natural Science Foundation of China (Grants #62025202) and the Frontier Technologies R&D Program of Jiangsu (BF2024059). Zhaoyu Li and Xujie Si were also supported, in part, by Individual Discovery Grants from the Natural Sciences and Engineering Research Council of Canada, and the Canada CIFAR AI Chair Program. Xian Zhang (`zhxian@microsoft.com`), Kaiyu Yang (`kaiyuy@meta.com`), and Xiaoxing Ma (`xxm@nju.edu.cn`) are the corresponding authors.

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

# A  SYMBOLIC METHODS VS. NEURAL METHODS

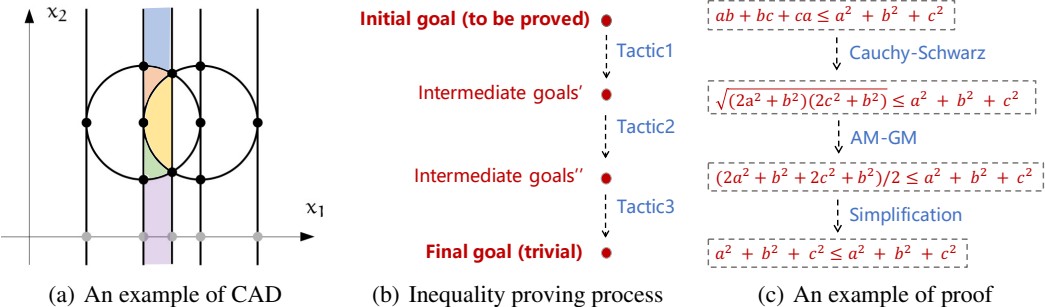

(a) An example of CAD        (b) Inequality proving process        (c) An example of proof

Figure 6: Figure (a) demonstrates how CAD is performed on two intersecting unit circles, deriving multiple sign-invariant cells (i.e., colored region). Figure (b) illustrates the process of inequality proving, which constructs a chain of proof goals by iteratively applying tactics; Figure (c) provides a corresponding instantiation of proving $ab + bc + ca \leq a^2 + b^2 + c^2$.

Figure 6(a) illustrates how symbolic methods work in proving inequalities. Symbolic methods are based on the CAD algorithm, which divides the underlying space $\mathbb{R}^n$ into multiple connected semi-algebraic sets. In each cell, the sign of every polynomial remains constant (positive, negative, or zero). Therefore, we just need to sample one point to determine the satisfiability of each cell, rather than scanning the whole $\mathbb{R}^n$ space. One can refer to Caviness & Johnson (2012); Arnon et al. (1984); Jirstrand (1995) for more details of CAD algorithm.

Figure 6(b) and 6(c) illustrate a commonly used paradigm in inequality proving. In a nutshell, a theorem prover often starts with the proof goal, and then iteratively transforms it into a simpler form until the final version can be easily confirmed. This approach has two main advantages. First, it ensures that the resulting proof can be more easily formalized in formal languages such as Lean. Second, since the hypotheses involved in inequality proofs are often straightforward, scaling and rewriting the proof goal is typically more efficient.

# B  PSEUDO CODE OF LIPS

Algorithm (3) outlines the overall process of LIPS proof generation. The detailed steps for tactic generation and pruning are provided in Algorithm 1, while the specifics for goal filtering and ranking are described in Algorithm 2.

---

**Algorithm 1** Tactic generation and pruning of LIPS

---

**Input:** A proof goal $g$; A lemma library of scaling tactics $\Phi$ and a prompt set of rewriting tactics $\Psi$; A language model $M$.
**Output:** Tactic set $T$.
 1: Initialize the tactic set $T = \{\}$
 2: **for** $t$ in $\Phi$ **do**                                                            ▷ *Scaling tactic generation*
 3:     Obtain arguments of the tactic $t$ using pattern_match on the goal $\Phi$.
 4:     Check the tactic $t$ (with derived arguments) via the symbolic solvers.
 5:     **if** no counterexamples exist **then**                          ▷ *Tactic pruning & Solver update*
 6:         Add the tactic $t$ into the tactic set $T$.
 7:     **else**
 8:         Update the symbolic solvers by including newly detected counterexamples.
 9:     **end if**
10: **end for**
11: **for** $t$ in $\Psi$ **do**                                    ▷ *Rewriting tactic generation & pruning*
12:     Obtain arguments of the tactic $t$ by prompting the LLM.
13:     Add the tactic $t$ (with derived arguments) into the tactic set $T$.
14: **end for**

---

---

**Algorithm 2** Goal filtering and ranking of LIPS

---

**Input:** A goal candidate set $\Omega$.
**Output:** A ranked set $\Omega'$.
  1: Initialize a new goal set $\Omega' = \{\}$.
  2: **for** $g$ in $\Omega$ **do**                                                   ▷ *Symbolic goal filtering*
  3:     Compute the homogeneity $\alpha$ and decoupling $\beta$ for $g$.
  4:     Define the average score of the goal $g$ by $(\alpha + \beta)/2$.
  5: **end for**
  6: Prompt the LLM to rank the first ten goals in $\Omega'$                         ▷ *Neural goal ranking*
  7: Obtain arguments of the tactic $t$ by prompting the LLM.
  8: Re-rank the goal set $\Omega'$ according to LLM's responses.

---

**Algorithm 3** Overall proof generation process of LIPS

---

**Input:** A formal inequality problem $g_0$; A lemma library of scaling tactics $\Phi$ and a prompt set of rewriting tactics $\Psi$; A language model $M$.
**Output:** A formal proof or timeout.
  1: Initialize the candidate goal set $\Omega = \{g_0\}$.
  2: **for** $i = 1, \ldots,$ **do**
  3:     Select the first goal $g$ in $S$ for exploration.
  4:     Obtain the tactic set $T$ by applying Algorithm (1) on the current goal $g$.
  5:     **for** $t$ in $T$ **do**                                                   ▷ *Tactic application*
  6:         Apply the tactic to the current $g$ in Lean, deriving a new goal $g'$.
  7:         Add the new goal $g'$ into the the candidate goal set $\Omega$.
  8:     **end for**
  9:     Update the goal set $\Omega$ by applying Algorithm (2).
 10: **end for**

---

## C    ADDITIONAL DETAILS FOR EXPERIMENTS

The experiments were conducted on four Linux servers equipped with 4 Intel(R) Xeon(R) Platinum 8280L CPU @ 2.80GHz. Each server ran Ubuntu 22.04 with GNU/Linux kernel 6.5.0-1015-azure. Each proving task was performed within a docker sandbox, utilizing 192 assigned CPU cores.

**Neural provers.**    For DSP, we directly use the official code, and adapt it to Lean language and GPT-4o (version Azure-05-01). MCTS is implemented based on classic upper confidence bounds applied to trees algorithm (Kocsis & Szepesvári, 2006). The value function is defined as $f(\phi) = v_\phi + C\sqrt{\log(N_\phi)/n_\phi}$, where $n_\phi$ is the number of the proof goal $\phi$ is selected and explored, $N_\phi$ represents the number of $\phi$'s parent represents selected and explored, and $C$ is a hyperparameter set to $C = \sqrt{2}$. For the average reward $v_\phi$ of the proof goal $\phi$, we use the same heuristic function as Wei et al. (2024), which calculates the maximum depth of the expression trees on both sides.

**Symbolic provers.**    For CAD, we utilize a portfolio including a suite of solvers, i.e., Z3, CVC5, RC-CAD, and Bottema. It will claim the problem is successfully proved if any one of four tools outputs `unsat`, and vice vica. Among four tools, Z3 and CVC5 are two popular SMT solvers; RC-CAD refers to the CylindricalAlgebraicDecompose function in Maple 2024 RegularChain package;[1] Bottema is a CAD-based inequality prover developed by Yang (1999).[2] As to MMA, we integrate two commands, i.e., `Reduce` and `FindInstance` in Wolfram-Mathematica (version 13.0.1), and apply the same peripheral logic with CAD.

**LIPS.**    In our framework, the symbolic solver employed for pruning scaling tactics also consists of solvers Z3, CVC5, RC-CAD, and Bottema, complemented by a numerical optimizer grounded in SciPy (Virtanen et al., 2020). The time limit of searching counterexamples is set to 5 seconds. Scaling tactics encompasses a comprehensive array of inequality lemma, including AM-GM, AM-HM, Cauchy-Schwarz, Power Mean, Chebyshev, Muirhead, Jensen, Titu, Schur, Holder inequalities,

---

[1] https://www.maplesoft.com/support/help/maple/view.aspx?path=RegularChains
[2] https://faculty.uestc.edu.cn/huangfangjian/en/article/167349/content/2378.htm

as well as a selection of valuable inequalities contributed by Lászió Kozma.[3] To facilitate their application in Lean 4, we have developed multiple variations of each inequality, accounting for different numbers of variables and directions. Figure 7 illustrates our scaling tactics using two-variable AM-GM inequality.

```
theorem NEQ_AM_GM_left_2vars (u v k l right : ℝ) (hk : k ≥ 0) (h : k *
    (u ^ 2 + v ^ 2) / 2 + l ≤ right) : k * (u * v) + l ≤ right := by
  suffices (u - v) ^ 2 ≥ 0 by nlinarith
  positivity

theorem NEQ_AM_GM_right_2vars (u v k l left : ℝ) (hk : k ≥ 0) (h : left
    ≤ 2 * k * (u * v) + l) : left ≤ k * (u ^ 2 + v ^ 2) + l := by
  suffices (u - v) ^ 2 ≥ 0 by nlinarith
  positivity
```

Figure 7: Two examples of AM-GM inequality encoded as scaling tactics

---

**Prompts of Rewriting Tactics Generation (Simplification)**

### Task
Your task is to use the condition {condition}, rearrange and rewrite the expression given by the user into an absolutely different form.
### Notice
1. Please reason step by step
2. Only four operators, add, sub, multiply, and division, can be used, and should NOT introduce new variables
3. Put the final results within \\boxed{{}}, e.g., \\boxed{{x + 1/y - z}}
### Response
User:
{problem}
Assistant:

---

**Prompts of Rewriting Tactics Generation (Others)**

### Task
You should rewrite the inequality given by the user according to the rule {rule}
### Notice
1. Please reason step by step
2. Follow the given example, and output the result for the given inequality
3. Put the final results within \\boxed{{}}, e.g., \\boxed{{x + 1/y - z}}
### Example
User:
{example_problem}
Assistant:
{example_answer}
### Response
User:
{problem}
Assistant:

Figure 8: Prompts of simplification and other operations used for generating rewriting tactics

For rewriting tactics, we design 16 relevant operations, i.e., simplification w/o assumptions, simplification w/ assumptions, completing the square, variable substitution, expression expansion, expression rearrangement, expression multiplication, cancellation of denominators/numerators, cancellation of powers, extraction/cancellation of common factors, separation/reduction of fractions, sum-of-squares trick, and tangent line trick. Except for the sum-of-squares trick and tangent line trick implemented based on SymPy, we use two prompt templates for these operations, shown in Figure 8. For each operation, we repeatedly query the LLM three times to cover more rewriting tac-

---

[3] https://www.lkozma.net/inequalities_cheat_sheet/ineq.pdf

tics. We use a low-temperature setting of GPT-4o in rewriting tactic generation and neural ranking (T=0.1, top_p=1.0, max_token=2048).

# D   ABLATION STUDY

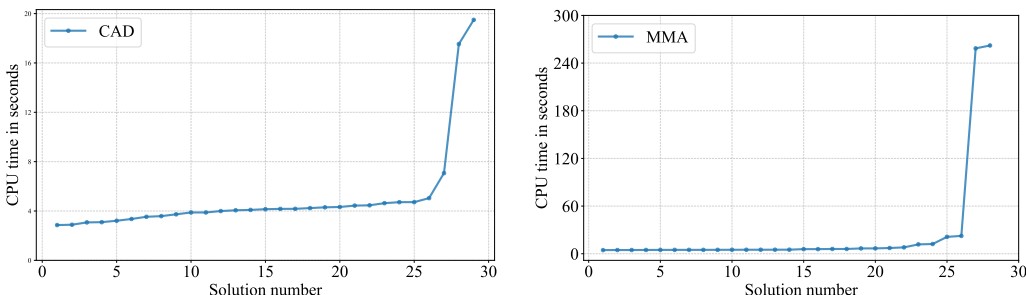

Figure 9: CPU time vs. solution count curves of CAD and MMA. The results show that symbolic solvers can verify the inequality in less than five minutes. For other problems that cannot be verified quickly, allocating more time does not improve the results. In addition, it is important to note that symbolic solvers are unable to construct human-readable proofs for the problems.

**Scaling tactics.** Figure 4(a) has shown how the performance on the ChenNEQ dataset (Y-axis) changes as we increase the number of lemmas used by scaling tactics (X-axis). When X = 0 (without scaling tactics), only 3 out of 41 theorems can be proved, whereas LIPS can prove 39 theorems.

**Rewriting tactics.** We observed that the successfully proved 3 theorems without scaling tactics are achieved by two rewriting tactics, sum-of-squares and tangent line trick. Hence, we further investigate the ablation of these two tactics. We newly evaluated LIPS w/o sum-of-squares and tangent lines on ChenNEQ. Among the 39 out of 41 proved inequalities, none of them requires sum-of-squares, and 3 requires the tangent line trick.

**Symbolic filtering.** Figure 4(b) has shown how the performance on the ChenNEQ dataset (Y-axis) changes as we increase the size of the filtered set (X-axis). A smaller X means more aggressive symbolic filtering. When X is large, e.g., 32, the performance drops under 40%, highlighting the importance of symbolic filtering. We did not test values of X beyond 32, as this consistently caused GPT-4o to hit the maximum token limit (set at 4096).

**Neural ranking.** Figure 5 presents an ablation analysis where neural ranking is replaced with random selection. The results on ChenNEQ demonstrate a significant drop in LIPS performance, from 95.1% to 41.2%, underscoring the necessity of neural ranking. Additionally, we incorporate a new ablation analysis that examines the effect of replacing neural ranking with purely symbolic filtering. The success rate decreased to 43.9%, indicating that symbolic filtering cannot determine an accurate proof path.

**Symbolic solvers.** To explore the performance of symbolic solvers, we use the ChenNEQ dataset and plot the CPU time vs. solution count curves on Figure 9. It can be observed that (1) symbolic solvers are very fast, they can verify the inequality in less than five minutes; (2) for other problems that cannot be verified quickly, allocating more time cannot improve the results. It also means that we only need to set a small time limit for symbolic solvers in LIPS.

Table 2: LIPS proving success rate if different time budgets

| Method | <10min | <30min | <50min | <70min | <90min |
|--------|--------|--------|--------|--------|--------|
| LIPS   | 5%     | 15%    | 35%    | 60%    | 80%    |

As to the results of LIPS, we list the success rate vs. CPU time in Table 2. Additionally, we plan to further enhance the efficiency of LIPS in the next version, for example, by optimizing the SymPy `pattern_matching` function.

# E   CASE STUDY

---

**A Failed Case of DSP**

**Prompt:** Draft an informal solution similar to below. The informal solution will be used to sketch a formal Lean proof. Here are some examples:

Example:

Informal:

(* ### Problem {Problem-1} ### Solution {Solution-1} *)

Informal:

(* ### Problem {Problem-2} ### Solution {Solution-2} *)

Informal:

(* ### Problem

if $a, b, c$ are positive real numbers, then

$$\frac{a}{b} + \frac{b}{c} + \frac{c}{a} \geq \sqrt{\frac{a^2+1}{b^2+1}} + \sqrt{\frac{b^2+1}{c^2+1}} + \sqrt{\frac{c^2+1}{a^2+1}}.$$

- - - - - - -

**Informal solution:**

### Solution

We will use the AM-GM inequality which states that for non-negative real numbers $x$ and $y$, the arithmetic mean is greater than or equal to the geometric mean: $\frac{x+y}{2} \geq \sqrt{xy}$.

First, let's consider the terms on the left-hand side one by one and apply AM-GM to each term along with its corresponding term on the right-hand side.

For the term $\frac{a}{b}$ and its corresponding term $\sqrt{\frac{a^2+1}{b^2+1}}$, we apply AM-GM to $\frac{a}{b}$ and $\frac{a}{b} \cdot \frac{1}{\frac{a^2+1}{b^2+1}} = \frac{b^2+1}{a}$:

$$\frac{\frac{a}{b} + \frac{b^2+1}{a}}{2} \geq \sqrt{\frac{a}{b} \cdot \frac{b^2+1}{a}} = \sqrt{\frac{a^2+1}{b^2+1}}.$$

Repeating this process for $\frac{b}{c}$ with $\sqrt{\frac{b^2+1}{c^2+1}}$ and $\frac{c}{a}$ with $\sqrt{\frac{c^2+1}{a^2+1}}$, we obtain:

$$\frac{\frac{b}{c} + \frac{c^2+1}{b}}{2} \geq \sqrt{\frac{b^2+1}{c^2+1}},$$

and

$$\frac{\frac{c}{a} + \frac{a^2+1}{c}}{2} \geq \sqrt{\frac{c^2+1}{a^2+1}}.$$

Adding these three inequalities together, we get:

$$\frac{1}{2}\left(\frac{a}{b} + \frac{b}{c} + \frac{c}{a} + \frac{b^2+1}{a} + \frac{c^2+1}{b} + \frac{a^2+1}{c}\right) \geq \sqrt{\frac{a^2+1}{b^2+1}} + \sqrt{\frac{b^2+1}{c^2+1}} + \sqrt{\frac{c^2+1}{a^2+1}}.$$

Notice that the left-hand side contains the terms $\frac{a}{b} + \frac{b}{c} + \frac{c}{a}$ and additional terms. Since $\frac{b^2+1}{a} + \frac{c^2+1}{b} + \frac{a^2+1}{c} > 0$ (because all terms are positive), we can drop these additional terms to strengthen the inequality:

$$\frac{a}{b} + \frac{b}{c} + \frac{c}{a} \geq \sqrt{\frac{a^2+1}{b^2+1}} + \sqrt{\frac{b^2+1}{c^2+1}} + \sqrt{\frac{c^2+1}{a^2+1}}.$$

Thus, we have shown that the original inequality holds for any positive real numbers $a, b, c$.*)

- - - - - - -

**Prompt:**

Translate the informal solution into a sketch of the formal Lean 4 proof. Add 'suggest_tactics' in the sketch whenever possible. 'suggest_tactics' will be used to call the automated theorem prover. Here are some examples:

Example:

Informal:

(* ### Problem

{informal_problem}

### Solution

{informal_solution}

Formal:
{formal_problem} *)
Informal:
(* ### Problem
{informal_problem}
### Solution
{informal_solution}
Formal:

- - - - - - - - - - - - - - - - - - - - - - - - - - - - - - - - - - - - - - - - - - - - - - - - -

**Formal solution:**

```
import Mathlib.Data.Real.Basic
import Mathlib.Tactic.Linarith
import Mathlib.Tactic.SuggestTactics

theorem inequality_amgm (a b c : ℝ) (ha : 0 < a) (hb : 0 < b) (hc
    : 0 < c) :
  (a / b) + (b / c) + (c / a) ≥
    (sqrt ((a^2 + 1) / (b^2 + 1))) + (sqrt ((b^2 + 1) / (c^2 +
    1))) + (sqrt ((c^2 + 1) / (a^2 + 1))) :=
by
  -- Apply the Arithmetic Mean-Geometric Mean Inequality (AM-GM)
  have am_gm_ab := real.am_gm (a / b) ((b^2 + 1) / a),
  have am_gm_bc := real.am_gm (b / c) ((c^2 + 1) / b),
  have am_gm_ca := real.am_gm (c / a) ((a^2 + 1) / c),

  -- The AM-GM inequality gives us the following:
  -- am_gm_ab : (a / b + (b^2 + 1) / a) / 2 ≥ sqrt ((a / b) *
    ((b^2 + 1) / a))
  -- am_gm_bc : (b / c + (c^2 + 1) / b) / 2 ≥ sqrt ((b / c) *
    ((c^2 + 1) / b))
  -- am_gm_ca : (c / a + (a^2 + 1) / c) / 2 ≥ sqrt ((c / a) *
    ((a^2 + 1) / c))

  -- Now, we sum the three inequalities:
  linarith [am_gm_ab, am_gm_bc, am_gm_ca],

  -- Simplify the right-hand side to match the terms of the
    original inequality
  -- 'suggest_tactics' could assist in automating algebraic
    simplifications here:
  suggest\_tactics,

  -- Conclude the proof using positivity of additional terms
  -- Positivity of terms like (b^2 + 1) / a and others should be
    straightforward
  have pos_terms : (b^2 + 1) / a + (c^2 + 1) / b + (a^2 + 1) / c >
    0,
  { apply add_pos, -- breaking it down into multiple positive parts
    all_goals { apply div_pos; linarith } },

  -- Dropping the additional positive terms from the left-hand
    side to strengthen the inequality
  linarith [pos_terms],
```

### Explanation of the Steps:
1) Initial Setup and Imports: We start by importing necessary modules from Mathlib, such as basic real number operations and the linarith tactic for linear arithmetic.
2) Statement of the Theorem: The theorem inequality_amgm is stated for three positive real numbers $a$, $b$, and $c$. The assumptions ha, hb, and hc enforce the positivity conditions.
3) ......
4) ......
### Conclusion
This sketch outlines the formal structure of the proof and highlights where automation can assist.

### A Failed Case of OpenAI o1-preview

**Prompt:** Please solve the following problem: For $a > 0, b > 0, c > 0, d > 0$ such that $a \cdot b + b \cdot c + c \cdot d + d \cdot a = 1$, show that:

$$\frac{a^3}{b+c+d} + \frac{b^3}{c+d+a} + \frac{c^3}{d+a+b} + \frac{d^3}{a+b+c} \geq \frac{1}{3}$$

- - - - - - - - - - - - - - - - - - - - - - - - - - - - - - - - - - - - - - - - - - - - - - -

**Informal solution:**
To solve this problem, we aim to show that for positive real numbers $a$, $b$, $c$, $d$ satisfying $ab + bc + cd + da = 1$, the inequality

$$\frac{a^3}{b+c+d} + \frac{b^3}{c+d+a} + \frac{c^3}{d+a+b} + \frac{d^3}{a+b+c} \geq \frac{1}{3}$$

holds true.

*Step 1: Simplify the Inequality.* First, note that

$$b + c + d = (a + b + c + d) - a = S - a,$$

where $S = a + b + c + d$. Thus, the inequality can be rewritten as:

$$\sum_{\text{cyclic}} \frac{a^3}{S - a} \geq \frac{1}{3}.$$

*Step 2: Analyze the Function.* Consider the function

$$f(x) = \frac{x^3}{S - x} \quad \text{for} \quad x \in (0, S).$$

We will investigate the convexity of this function to apply Jensen's inequality.
Compute the second derivative:

$$f''(x) = \frac{2S^2 + 2Sx + x^2}{(S - x)^3} > 0 \quad \text{for} \quad x \in (0, S).$$

This shows that $f(x)$ is convex on $(0, S)$.
*Step 3: Apply Jensen's Inequality.* Since $f(x)$ is convex, Jensen's inequality states that:

$$\frac{1}{4} \sum_{i=1}^{4} f(a_i) \geq f\left(\frac{1}{4} \sum_{i=1}^{4} a_i\right) = f\left(\frac{S}{4}\right)$$

Compute $f\left(\frac{S}{4}\right)$:

$$f\left(\frac{S}{4}\right) = \frac{\left(\frac{S}{4}\right)^3}{S - \frac{S}{4}} = \frac{\left(\frac{S}{4}\right)^3}{\frac{3S}{4}} = \frac{S^2}{48}$$

Therefore, the sum becomes:

$$\sum_{\text{cyclic}} \frac{a^3}{S - a} \geq 4 \cdot \frac{S^2}{48} = \frac{S^2}{12}$$

*Step 4: Relate S to the Given Condition.* From the given condition $ab + bc + cd + da = 1$, and using the Cauchy-Schwarz inequality:

$$(ab + bc + cd + da) \leq \frac{1}{2}\left(a^2 + b^2 + c^2 + d^2\right)$$

But since $ab + bc + cd + da = 1$, we get:

$$a^2 + b^2 + c^2 + d^2 \geq 2$$

Also, note that

$$\color{red} S^2 = (a + b + c + d)^2 \geq 4(a^2 + b^2 + c^2 + d^2) \geq 8, \quad \text{so } S^2 \geq 8.$$

Therefore:

$$\color{red} \sum_{\text{cyclic}} \frac{a^3}{S - a} \geq \frac{S^2}{12} \geq \frac{8}{12} = \frac{2}{3}$$

But this contradicts the desired inequality, suggesting a miscalculation.

*Step 5: Test the Equality Case.* Consider $a = b = c = d = \frac{1}{2}$. Then:

$$ab + bc + cd + da = 4\left(\frac{1}{2} \cdot \frac{1}{2}\right) = 1$$

Compute the left-hand side:

$$\sum_{\text{cyclic}} \frac{a^3}{b+c+d} = 4 \cdot \frac{\left(\frac{1}{2}\right)^3}{\frac{3}{2}} = 4 \cdot \frac{\frac{1}{8}}{\frac{3}{2}} = 4 \cdot \frac{1}{8} \cdot \frac{2}{3} = \frac{1}{3}$$

This shows that equality holds when $a = b = c = d = \frac{1}{2}$.

*Step 6: Conclude the Proof* Since the function $f(x) = \frac{x^3}{S-x}$ is convex, the minimum of the sum occurs when all variables are equal, i.e., $a = b = c = d$. Therefore, the inequality holds true:

$$\frac{a^3}{b+c+d} + \frac{b^3}{c+d+a} + \frac{c^3}{d+a+b} + \frac{d^3}{a+b+c} \geq \frac{1}{3}.$$

---

## A Failed Case of LIPS

**Problem (USAMO 2004):** Let $a, b, c$ be positive reals. Prove that

$$(a + b + c)^3 \leq (a^5 - a^2 + 3) * (b^5 - b^2 + 3) * (c^5 - c^2 + 3)$$

- - - - - - - - - - - - - - - - - - - - - - - - - - - - - - - - - - - - - - - - - - - - - - - - - - - - - - - -

**Formal solution:**

```
theorem Problem_3dP5 {a b c : ℝ} (ha : a > 0) (hb : b > 0) (hc : c
    > 0) : (a + b + c) ^ 3 ≤ (a ^ 5 - a ^ 2 + 3) * (b ^ 5 - b ^ 2 +
    3) * (c ^ 5 - c ^ 2 + 3) := by
  scale NEQ_Cauchy_Schwarz_left_sqrt_3vars (u1 := 1) (u2 := 1) (u3
    := 1) (v1 := a) (v2 := b) (v3 := c) ...
  scale NEQ_weighted_AM_GM_left_3vars (u := a + b + c) (v := 3) (w
    := a^2 + b^2 + c^2) (i1 := 1) (i2 := 1/2) (i3 := 1/2) ...
  llm_frac_reduce ...
  llm_cancel_denom ...
  scale NEQ_Cauchy_Schwarz_sqrt_left_2vars (u1 := 1) (u2 := 1) (v1
    := a) (v2 := a) ...
  llm_simplify ...
  llm_rearrange ...
  llm_simplify ...
  llm_rearrange ...
  llm_simplify ...
  llm_rearrange ...
  llm_frac_apart ...
  llm_simplify ...
  llm_rearrange ...
  llm_frac_apart ...
  llm_simplify ...
  llm_simplify ...
  llm_rearrange ...
  llm_simplify ...
  llm_rearrange ...
  llm_simplify ...
  llm_factor ... = 4*sqrt (2)*sqrt (b^2 + c^2)*(3 + a^2 + b^2 + c^2
    + 2*a + 2*b + 2*c)^2 + (1 + a)^2*(3 + a^2 + b^2 + c^2 + 2*a +
    2*b + 2*c)^2 - (64*(3 + a^5 - a^2)*(3 + b^5 - b^2)*(3 + c^5 -
    c^2))
  FAIL (TIMEOUT) !
```

---

**A Successful Case of LIPS**

**Problem (Evan Chen):** If $a, b, c$ are positive reals and $a^2 + b^2 + c^2 = 1$, then

$$\frac{1}{a^2 + 2} + \frac{1}{b^2 + 2} + \frac{1}{c^2 + 2} \leq \frac{1}{6ab + c^2} + \frac{1}{6bc + a^2} + \frac{1}{6ca + b^2}.$$

---

**Formal solution:**

```
theorem Problem_2d4e1 {a b c : ℝ} (ha : a > 0) (hb : b > 0) (hc :
    c > 0) (h : a ^ 2 + b ^ 2 + c ^ 2 = 1) : 1 / (a ^ 2 + 2) + 1 /
    (b ^ 2 + 2) + 1 / (c ^ 2 + 2) ≤ 1 / (6 * a * b + c ^ 2) + 1 /
    (6 * b * c + a ^ 2) + 1 / (6 * c * a + b ^ 2) := by
  scale NEQ_AM_GM_div_square_right_2vars (u := b) (v := c) ...
  scale NEQ_AM_GM_div_square_right_2vars (u := b) (v := a) ...
  scale NEQ_AM_GM_div_square_right_2vars (u := a) (v := c) ...
  llm_simplify ...
  llm_simplify ...
  llm_rearrange ...
  sym_tangent_line ...
  llm_simplify ... = 27/49 - (27*a^2/49) - (27*b^2/49) - (27*c^2/49)
  try close
  SUCCESS!
```

## F   SOME MORE CASES OF LIPS

---

**A Different Proof Generated by LIPS (1)**

**Problem:** Let $a, b, c$ be three positive reals. Prove that if $abc = 1$, then

$$a^2 + b^2 + c^2 \geq a + b + c$$

Source: `https://web.evanchen.cc/handouts/Ineq/en.pdf`

---

**Formal solution:**

```
theorem Example_1d7 {a b c : ℝ} (h : a * b * c = 1) : a + b + c ≤
    a ^ 2 + b ^ 2 + c ^ 2 := by
  scale NEQ_AM_GM_left_square_2vars (u := 1) (v := a) ...
  scale NEQ_AM_GM_left_square_2vars (u := 1) (v := c) ...
  scale NEQ_AM_GM_left_square_2vars (u := 1) (v := b) ...
  llm_rearrange ...
  llm_simplify ...  = 3/2 - a^2/2 - b^2/2 - c^2/2
  llm_rearrange (left := 3/2) (right := a^2/2 + b^2/2 + c^2/2)
  scale NEQ_AM_GM_right_normal_3vars (u := a^2/2) (v := b^2/2) ...
  llm_simplify ...  = (a*b*c)^2 / 8
  llm_simplify ...  = 1 / 8
  try close
```

Note: At first glance, "AM-GM alone is hopeless here, because whenever we apply AM-GM, the left and right hand sides of the inequality all have the same degree". However, LIPS find a proof by using $1 \times a \leq \frac{a^2+1}{2}$, $1 \times b \leq \frac{b^2+1}{2}$, $1 \times c \leq \frac{c^2+1}{2}$.

---

### A Different Proof Generated by LIPS (2)

**Problem (JMO 2012):** Let $a, b, c$ be positive reals. Prove that

$$\frac{a^3 + 5b^3}{3a + b} + \frac{b^3 + 5c^3}{3b + c} + \frac{c^3 + 5a^3}{3c + a} \geq \frac{3}{2}(a^2 + b^2 + c^2).$$

Source: https://artofproblemsolving.com/community/c5h476722p2669114

- - - - - - - - - - - - - - - - - - - - - - - - - - - - - - - - - - - - - - - - - - - - - - - - - - - -

**Formal solution:**

```
theorem Problem_3de6'' {a b c : ℝ} (ha : a > 0) (hb : b > 0) (hc :
    c > 0) : (3 / 2) * (a ^ 2 + b ^ 2 + c ^ 2) ≤ (a ^ 3 + 5 * b ^
    3) / (3 * a + b) + (b ^ 3 + 5 * c ^ 3) / (3 * b + c) + (c ^ 3 +
    5 * a ^ 3) / (3 * c + a) := by
  llm_cancel_factor ...
  scale NEQ_Muirhead2_onestep_right_3vars (u := a) (v := b) ...
  scale NEQ_Muirhead1_left_3vars (u := c) (v := a) (w := b) ...
  scale NEQ_Muirhead1_left_3vars (u := a) (v := b) (w := c) ...
  scale NEQ_Muirhead2_onestep_left_3vars (u := a) (v := b) ...
  llm_cancel_factor ...
  scale NEQ_Muirhead2_onestep_right_3vars (u := a) (v := b) ...
  llm_simplify ...
  scale NEQ_Muirhead2_left_3vars (u := c) (v := b) (w := a) ...
```

Note: A completely mechanical proof, achieved by iteratively applying Muirhead inequality six times after canceling the denominator

---

### LIPS Checks Existing Answers (1)

**Problem (Japan 1997):** Prove that

$$\frac{(b + c - a)^2}{(b + c)^2 + a^2} + \frac{(c + a - b)^2}{(c + a)^2 + b^2} + \frac{(a + b - c)^2}{(a + b)^2 + c^2} \geq \frac{3}{5}$$

Source: https://artofproblemsolving.com/community/c6h146p537

- - - - - - - - - - - - - - - - - - - - - - - - - - - - - - - - - - - - - - - - - - - - - - - - - - - -

**Informal solution No.1 (provided by AoPS community):**
Solution:
Let :
$P = \frac{(b+c-a)^2}{(b+c)^2+a^2} + \frac{(c+a-b)^2}{(c+a)^2+b^2} + \frac{(a+b-c)^2}{(a+b)^2+c^2}$
By Cauchy Swarchz inequality, we have

$$(a^2 + b^2 + c^2 + (a + b)^2 + (b + c)^2 + (c + a)^2)(P) \geq (a + b + c)^2$$
$$(3(a^2 + b^2 + c^2) + 2(ab + bc + ca))(P) \geq (a + b + c)^2$$
$$(3(a + b + c)^2 - 4(ab + bc + ca))(P) \geq (a + b + c)^2$$

Hence,
$P \geq \frac{(a+b+c)^2}{3(a+b+c)^2-4(ab+bc+ca)^2}$
but we know that $(a + b + c)^2 \geq 3(ab + bc + ca)$,
$P \geq \frac{3(ab+bc+ca)}{9(ab+bc+ca)-4(ab+bc+ca)} = \frac{3}{5}$
...

Note: the proof is correct if all variables are assumed to be positive. However, without this assumption, applying Cauchy-Schwarz is incorrect, and a counterexample is successfully found by our symbolic solver $[c := 8.0, a := 1/8, b := (-1.0)]$.
This problem is proved by LIPS using llm_simplification and tangent_line trick tactics.

---

**LIPS Checks Existing Answers (2)**

**Problem (Japan 1997):** Prove that

$$\frac{(b+c-a)^2}{(b+c)^2+a^2} + \frac{(c+a-b)^2}{(c+a)^2+b^2} + \frac{(a+b-c)^2}{(a+b)^2+c^2} \geq \frac{3}{5}$$

Source: `https://artofproblemsolving.com/community/c6h146p537`

- - - - - - - - - - - - - - - - - - - - - - - - - - - - - - - - - - - - - - - - - - - - - - - -

**Informal solution No.2 (provided by AoPS community):**

....
don't use so much of mathematics on that simple problem..
:P :D here's the solution using TITU's lemma..
The lemma-
if $x, y, a, b$ are reals and $x, y > 0$ then $\frac{a^2}{x} + \frac{b^2}{y} \geq \frac{(a+b^2}{(x+y)}$
simply applying the lemma on the lhs twice
we get
$LHS \geq \frac{(a+b+c)^2}{3(a^2+b^2+c^2)+2(ab+bc+ca)}$
by AM-GM
$(a+b+c)^2 \geq 3(ab+bc+ca)$ and $3(a^2+b^2+c^2)+2(ab+bc+ca) \geq 5(ab+bc+ca)$
...
Note: it appears that the issue arises from the inequality being in the wrong direction. Actually, the problem is unprovable when applying the Titu's lemma (a counterexample found by our symbolic solver: $[b=1.0, a=1.0, c=0.5]$)

