# OpenReview forum: "Proving Olympiad Inequalities by Synergizing LLMs and Symbolic Reasoning"
_ICLR.cc/2025/Conference — ICLR 2025 Poster_

### Official Review · Reviewer_YrrG · 2024-10-29

**Soundness:** 3
**Presentation:** 4
**Contribution:** 2
**Rating:** 6
**Confidence:** 2

**Summary:**

The paper presents a neuro-symbolic tactic generator designed to solve typical inequality problems found in
maths olympiad contests. The authors identify two general strategies often employed by humans in solving these problems, which involve creating new goals via:
- Scaling, which involves the application of known lemmas, inequalities, etc., to the inequality that needs solving.
- Rewriting, which involves using common equivalent transformations on the inequality that needs solving.

Lips utilises a blend of symbolic methods, used for scaling and some rewriting, and LLMs, primarily used for rewriting inequalities and ranking and selecting goals. The performance of Lips is compared with state of the art tools on common benchmarks. Additionally, general paradigms for neuro-symbolic reasoning are extracted from the approach Lips takes for math olympiad inequality problems.

**Strengths:**

- The development of neuro-symbolic tools for reasoning about formal tasks is quite intriguing, and this paper contributes nicely in that its findings seem to transcend the specific task of solving maths olympiad questions.
- The approach taken by Lips is explained in detail and is easy to comprehend. Overall, the concept behind Lips appears sound.
- The experimental parts of the paper are sufficiently supported.
- The paper is well-written and organized, with everything supported by examples.
I found it an enjoyable read.

**Weaknesses:**

I order the weaknesses from most to least important:
- It is unclear to me how impactful the findings of this paper are beyond
the development of a well-performing neuro-symbolic tactic generator
for these specific inequality problems. While the workflow of Lips
is a nice combination of symbolic and LLM-based components, the underlying
idea seems rather straightforward: whenever symbolic methods are
sufficient (as in the scaling steps), use symbolic; and whenever things
become less clear (as in rewriting and goal selection), use LLM. While I
may be incorrect, this seems folklore to me.
- While I appreciate the paper's lightweight tone, at times,
it feels that many choices are justified by intuition alone.
For example, the authors state that scaling and rewriting are the methods humans use to solve such inequality problems. This seems reasonable to me, yet there is no evidential support for this in the paper. Similarly, in Section 4.1, the authors introduce criteria for ranking proof goals.
Again, these appear reasonable to me, but there is no support for these claims.
It may well be the case that research on such questions is simply non-existent, but I would have appreciated it being stated more clearly why we have to rely on intuition.
- At some points, the paper's language is a bit too informal. For example,
in the paragraph starting with line 180, it is stated that OpenAI's o1-preview
"cannot solve a specific inequality". The proof presented is an example
of a prompt given to o1-preview, where it fails to address this particular
inequality. Clearly, this is not a sufficient argument to claim that
o1-preview is generally unable to solve such inequality tasks.

**Questions:**

- As mentioned in my weaknesses, I remain unconvinced about the broader impact of this paper beyond
the neuro-symbolic resolution of inequality problems. Could you clarify what the novel findings are
in a wider context? Maybe even what are specific tasks, problems that benefit from your reasearch, aside
math olympiad problems?

- The development of a tool like Lips could be noteworthy, but I am not familiar with similar existing
tools. Am I correct in understanding that it is the first tool to integrate neural and symbolic components?
In your experiments, it appears that you only compare it to tools that are either purely neural or purely
symbolic.

---

> ### Author Response · Authors · 2024-11-18
> **Rebuttal by Authors**
>
> **Dear reviewer YrrG:**
>
> Thank you for taking the time to review our paper and provide feedback! Below we address your questions and concerns. Please feel free to follow up if you have further questions!
>
> **1. Justification for Scaling/Rewriting Tactics and Heuristics for Filtering Proof Goals**
>
> The intuition is largely drawn from notes and books written by IMO coaches. For scaling/rewriting tactics, Bruce Ikenaga summarizes in his note on inequality that ``You can often prove an inequality by transforming or substituting in a known inequality'' ([source](https://sites.millersville.edu/bikenaga/math-proof/inequalities/inequalities.html)). Similar discussions can also be found on [MathExchange](https://math.stackexchange.com/questions/4700745/question-about-how-to-prove-a-inequality). Regarding goal ranking, concepts such as coupling and homogeneity are frequently emphasized in references like [1, 2], which is why we incorporate them as metrics. We are unaware of existing literature studying these intuitions systematically.
>
> [1] Mildorf, Thomas J. "Olympiad Inequalities." Preprint, https://artofproblemsolving.com/articles/files/MildorfInequalities.pdf
>
> [2] Evan Chen. "A brief introduction to olympiad inequalities." Preprint, https://web.evanchen.cc/handouts/Ineq/en.pdf
>
> **2. Evaluating OpenAI o1**
>
> The line starting at Line 180 only claims o1 cannot solve the particular inequality problem in Figure 1. This paragraph does not claim o1 cannot solve a class of problems. However, we also evaluated o1 on the MO-INT dataset, and it wasn’t able to solve any of the problems (Line 398 and Appendix D). In contrast, LIPS solved 16 out of 20 problems in MO-INT (Table 1).
>
> **3. Comparisons with Existing Tools for Proving Inequalities**
>
> Thank you for bringing up an excellent question! To the best of our knowledge, LIPS is the first tool to integrate neural and symbolic reasoning for both tactic generation and subgoal evaluation, providing a comprehensive approach to inequality theorem proving.
>
> Existing tools generally fall into one of two categories: (1) purely symbolic methods, such as CAD solvers, which do not produce interpretable proofs, or (2) neural-based methods, like INT [1], that rely entirely on neural networks for tactic generation and subgoal evaluation. Among these, AIPS [2] takes a slightly different approach by leveraging symbolic methods for tactic generation and training a language model for subgoal evaluation.
>
> In contrast, LIPS conducts both tactic generation and goal ranking in a neuro-symbolic manner. Specifically, while AIPS relies on a fixed set of symbolic rewritings, LIPS particularly uses LLMs to handle rewriting tactics, allowing for a more flexible and dynamic action space, which better reflects the nature of mathematical reasoning.
> For goal ranking, AIPS synthesizes similar inequalities and fine-tunes an LLM to rank the subgoals, whereas LIPS combines symbolic filtering with ranking by GPT-4. Additionally, AIPS assumes the inequality problem is cyclically symmetric and tight, whereas our neuro-symbolic framework steps aside from these assumptions, providing greater generality. As a result, the neuro-symbolic design of LIPS enables it to significantly outperform AIPS in empirical evaluations.
>
> [1] Wu, Yuhuai, et al. "Int: An inequality benchmark for evaluating generalization in theorem proving." ICLR 2021
>
> [2] Wei, Chenrui, Mengzhou Sun, and Wei Wang. "Proving Olympiad Algebraic Inequalities without Human Demonstrations." NeurIPS Datasets and Benchmarks Track 2024
>
> **4. Suggestions on the Writing**
>
> Thank you for pointing out potential confusions! We will clarify them using more precise language in the next revision.

---

> > ### Comment · Reviewer_YrrG · 2024-11-19
> >
> > Thank you for your response.
> >
> > > Justification for Scaling/Rewriting Tactics and Heuristics for Filtering Proof Goals
> >
> > Thank you for providing additional clarification. I believe it would be beneficial to incorporate at least an informal paragraph summarising the justifications you provided above. I feel this would only strengthen your results.
> >
> > > The line starting at Line 180 only claims o1 cannot solve the particular inequality problem in Figure 1.
> >
> > That is how I understood it, but the claim does not appear to be clear and seems incorrect to me. This is due to a lack of formally defining what "solving a problem" means. If I am not mistaken, you provided a single prompt for which o1-preview failed to solve the inequality problem of Figure 1. However, perhaps a slightly different prompt would suffice?
> >
> > I believe I have an idea of your perspective. You always provide the problem as a prompt in the exact form it would appear in an Olympiad, correct? Then, a model $M$ (for example, o1-preview) solves an inequality problem $P$ if, when given the "Olympiad-description" of $P$ as a prompt, it "provides the correct proof". I think you understand what I am implying.
> >
> > Therefore, my question is: Am I correct in this understanding, and if so, is there a rigorous definition of "Olympiad-description"?
> > - If yes, did you make this clear in the paper?
> > Please feel free to direct me to the corresponding section.
> > - If no, I believe such a definition would assist individuals who are less familiar with this topic.
> >
> > > Comparisons with Existing Tools for Proving Inequalities
> >
> > Thank you for clarifying this!
> >
> > Concerning your general comment: You mentioned a possible connection to SMT solvers, which I perceive. However, I would expect that neuro-symbolic approaches are already being investigated in these contexts. Do you have a more detailed insight into what novel contributions LIPS could bring to this area of research?

---

> ### Author Response · Authors · 2024-11-20
>
> Thank you for following up! Below is our detailed response.
>
> > beneficial to incorporate at least an informal paragraph summarising the justifications you provided above
>
> Thank you for your suggestion! We will incorporate the justifications into our paper.
>
>
> > you provided a single prompt for which o1-preview failed to solve the inequality problem of Figure 1. However, perhaps a slightly different prompt would suffice?.
>
>
> Thank you for the clarification. We apologize if the original statement had caused confusion. To make the claim more precise, we will re-word it along the lines of "o1 cannot solve the inequality in Figure 1 when the problem is presented directly in the prompt".
>
> Our prompt for o1 is
> ```
> Please solve the following problem:
> <PROBLEM_STATEMENT>
> ```
> where `<PROBLEM_STATEMENT>` is the original problem statement in LaTex from the problem's original source, e.g., Chen [1] and Tung [2]. We do not define "Olympiad-description" or try to normalize the problem statement into a canonical format.
>
> As recommended by [OpenAI's official o1 prompting guidelines](https://platform.openai.com/docs/guides/reasoning#advice-on-prompting), a simple and direct prompt like ours will likely outperform more sophisticated prompts that employ techniques such as few-shot demonstrations or chain-of-thought prompting. Similarly straightforward prompts have been used for evaluating o1 in recent studies [3, 4].
>
> Additionally, we newly experimented with several different prompts, including:
> * Chain-of-thought prompt (“Let’s think step-by-step”)
> * System prompt (“You are an experienced educator in the field of MATHEMATICS”) [5]
> * A popular prompt shared on [Reddit](https://www.reddit.com/r/ChatGPTPromptGenius/comments/1fmngpu/try_gpt_o1_without_premuim_using_this_prompt/).
>
> Despite these efforts, the o1-preview model still fails to solve this particular inequality problem.
>
> [1] Chen. "A brief introduction to olympiad inequalities." 2014
>
> [2] Tung. "567 Nice And Hard Inequalities." 2012
>
> [3] Li et al. "OpenAI-o1 AB Testing: Does the o1 model really do good reasoning in math problem solving?." arXiv 2024
>
> [4] Zhong et al. "Evaluation of OpenAI o1: Opportunities and challenges of AGI." arXiv 2024
>
> [5] Gao et al. "Omni-math: A universal olympiad level mathematic benchmark for large language models." arXiv 2024
>
>
>
> > possible connection to SMT solvers … I would expect that neuro-symbolic approaches are already being investigated in these contexts
>
> To the best of our knowledge, neuro-symbolic approaches have not been used to enhance SMT solvers in the specific context discussed in the common response.
>
> SMT solvers are complex systems with many components (see Figure 1 of [1] for an overall architecture). Broadly speaking, an SMT solver integrates various decision procedures (e.g., for integers, strings, and real numbers) and uses them effectively to solve problems. Existing work has attempted to use machine learning to improve SMT from many directions, e.g., synthesizing custom Z3 strategies [2, 3], learning heuristics for quantifier instantiation [4, 5], and selecting the best solver for a particular problem from multiple solvers [6]. However, these approaches generally focus on how decision procedures are applied. They operate at a higher, meta-level, treating the decision procedures as black boxes.
>
> In contrast, what we proposed in the common response was to potentially use LIPS to replace or enhance the decision procedure for quantifier-free non-linear real arithmetic. A related line of work uses machine learning to learn heuristics in cylindrical algebraic decomposition (CAD) [7, 8]. CAD differs from LIPS (inner workings; whether human-readable proofs can be generated; discussed in Sec. 2.1). Nevertheless, one can potentially extract intermediate inequalities from LIPS’ proof search process as data for training machine learning models to enhance CAD.
>
> [1] Bjørner et al. "Programming Z3." Engineering Trustworthy Software Systems: 4th International School, SETSS 2018
>
> [2] Balunovic et al. "Learning to solve SMT formulas." NeurIPS 2018
>
> [3] Lu. “AlphaSMT: A Reinforcement Learning Guided SMT Solver”. University of Waterloo, 2023.
>
> [4] Janota et al. "Towards learning quantifier instantiation in SMT." SAT 2022
>
> [5] Blanchette et al. "Machine learning for instance selection in SMT solving." AITP 2019
>
> [6] Scott et al. "MachSMT: A machine learning-based algorithm selector for SMT solvers." International Conference on Tools and Algorithms for the Construction and Analysis of Systems. 2021.
>
> [7] Huang et al. "Using machine learning to improve cylindrical algebraic decomposition." Mathematics in Computer Science. 2019
>
> [8] Jia et al. "Suggesting variable order for cylindrical algebraic decomposition via reinforcement learning." NeurIPS 2024

---

> ### Comment · Reviewer_YrrG · 2024-11-21
>
> I see where you are coming from regarding the potential benefits for SMT solving, however our discussion about this remains superficial, which is partly due to the fact that I have not much background knowledge about the existing literature.
>
> I feel my issues mostly resolved and I do not feel that I can contribute anything of worth to the rebuttal anymore.
> I am not fully sure about the broader impact of this work, but I acknowledge its contribution.
>
> Accordingly, I increased my rating to a "weak accept" as I think its an overall nice contribution, but I decreased my confidence to "2" as it became clear that I could very likely be not aware of important related work.
>
> Thanks for the nice discussion, I really appreciate the way you are handling the rebuttal!

---

> > ### Author Response · Authors · 2024-11-24
> >
> > Thank you! It was great to discuss with you and other reviewers. Your feedback has been extremely helpful for us to clarify our thought process and improve the paper!

---

### Official Review · Reviewer_v6PZ · 2024-10-30

**Soundness:** 4
**Presentation:** 4
**Contribution:** 4
**Rating:** 8
**Confidence:** 3

**Summary:**

The paper introduces LIPS, a neuro-symbolic system that combines large language models (LLMs) and symbolic reasoning to solve Olympiad-level inequality problems in formal theorem proving.

LIPS divides tactics into scaling (using symbolic tools) and rewriting (using LLMs):

Scaling applies known inequality lemmas (e.g., AM-GM or Cauchy-Schwarz inequalities) to refine subterms in the target inequality. The symbolic approach allows exhaustive enumeration of lemma applications across potential arguments. However, only certain applications lead to correct deductions since overscaling risks invalidating the inequality. To address this, LIPS uses counterexample filtering to eliminate incorrect transformations. This involves generating possible solutions with symbolic tools (e.g., CAD solvers) to check whether the inequality holds across variable values, discarding transformations that fail.

Rewriting tactics focus on algebraic transformations to express terms in alternative but equivalent forms. This space is vast (infinite transformations possible), so LLMs are employed to guide the selection, generating transformations that preserve provability. LIPS designs prompts for common transformations (e.g., factorization, denominator simplification), allowing the LLM to filter likely useful transformations.

The authors conduct evaluations on 161 challenging inequality problems and demonstrate their system is significantly more capable than existing systems.

**Strengths:**

The paper takes a novel approach (to my knowledge) to solve olympiad inequalities following the recent paradigm shift of using test-time compute to solve mathematical problems. The domain specificity of solving inequality problems is very well exploited. The method rigorously categorizes proof tactics into scaling and rewriting, with each category handled by techniques suited to its complexity. This results in successful results as demonstrated by extensive experiments and results in proofs written in Lean 4 and hence has guarantees.

LIPS sets a new benchmark for inequality theorem proving by combining machine learning with symbolic techniques, which is promising for broader applications in automated mathematics.

The paper is well-organized, clearly explaining both the technical details of scaling and rewriting, and the hybrid neuro-symbolic approach.

**Weaknesses:**

It is unclear to me how LLM rewritings are formalized in Lean 4 - is it a Lean 4 tactic that can do the rewrites of LLM? If that is the case, it would be interesting to use this as a tactic in general theorem proving in Lean 4.

The datasets used (e.g., ChenNEQ, MO-INT, 567NEQ) focus on specific competition-level inequalities, which might limit generalizability. I don't understand why authors took a random sample of 100 from 567NEQ instead of running the system on all of them. I think it is important to make sure that there was no contamination due to solutions of these problems appearing in training data of GPT4o, although I believe the failure of other LLM based approaches is evidence that LISP's added value is significant.

I believe some evaluation comparing to existing theorem provers such as the one from DeepSeek or LeanCopilot could be included. I would also love to see the code on how the LLM rewriting tactics could be formalized in Lean 4.

**Questions:**

- Can you describe how the Lean 4 commands such as `llm_simplify`, `llm_rearrange`, etc. do? It would be great if you could provide a complete Lean 4 code that people can run for reproducibility and verification purposes.
- It would be interesting also to compare to existing Neural Theorem Provers in Lean 4 such as DeepSeek Prover V1.5 or LeanCopilot.

---

> ### Author Response · Authors · 2024-11-18
> **Rebuttal by Authors**
>
> **Dear reviewer v6PZ:**
>
> Thank you for taking the time to review our paper and provide feedback! Below we address your questions and concerns. Please feel free to follow up if you have further questions!
>
> **1. Implementation Details of Rewriting Tactics in Lean 4**
>
> Please see our common response at the top of this page.
>
> **2. Why Using a Subset of 100 Problems from 567NEQ?**
>
> We selected a subset of 567NEQ primarily due to budget constraints. However, we plan to extend our evaluation of LIPS to the full dataset in future work.
>
> **3. Potential Data Contamination Due to GPT-4o**
>
> Please see our common response at the top of this page.
>
> **4. Comparisons with DeepSeek-Prover-V1.5 and Lean Copilot**
>
> As suggested by the reviewer, we evaluated DeepSeek Prover V1.5 and LeanCopilot on the Mo-INT dataset. For LeanCopilot, we used the default setting and apply the  `search_proof` tactic. We observe that LeanCopilot fails to prove any problems of this challenging dataset, consistent with the result reported in AIPS. For DeepSeek Prover V1.5, we used the  DeepSeek-Prover-V1.5-RL model provided on [Huggingface](https://huggingface.co/deepseek-ai/DeepSeek-Prover-V1.5-RL) and evaluated its performance using the Pass@32 metric. Similarly, it also fails to solve any of the 20 problems.

---

### Official Review · Reviewer_cEd1 · 2024-11-03

**Soundness:** 2
**Presentation:** 3
**Contribution:** 2
**Rating:** 5
**Confidence:** 4

**Summary:**

The authors introduce LIPS: a hybrid system for proving inequalities, which combines symbolic solvers, manually crafted heuristics, and   LLMs. LIPS operates within the formal framework of Lean 4. The system is evaluated on a test set of 161 competition-style inequalities of varying difficulty, and is compared to 5 other methods (2 purely symbolic and 3 ML-based). LIPS performs significantly better than the baselines on the considered test set given the time limit of 90 minutes.

**Strengths:**

The work demonstrates the potential of combining machine learning and symbolic/formal techniques, which is an interesting and still under-explored direction.

The design of LIPS looks well thought out, leveraging appropriate symbolic techniques in the pipeline.

The authors compared LIPS against a representative variety of baselines, both symbolic and ML-based.

The paper is mostly written clearly, and provides enough background information and helpful examples (but some parts are less clear -- see weaknesses below).

**Weaknesses:**

The test dataset is not publicly available. In particular, the Lean and SMT formalizations are not available. It is relatively easily to introduce unintended errors when formalizing math, so open-sourcing the formalizations is important to make inspection possible. Also, it is crucial to allow other researchers to compare their approaches by providing the test data.

The implementation of LIPS and MCTS is not available for testing and inspection, and there is no promise of open sourcing it.

Some parts of the description of LIPS are unclear to me -- see questions below. In particular, it is described how LIPS generate a set of new proof goals from one proof goal, but the main body of the paper does not contain any information on how search is conducted in the resulting tree of proof goals.

The related work section is missing important related literature. Combining formal theorem proving with machine learning is not limited to LLMs. There is a rich body of research combining other ML approaches and proof assistants [1,2,3,4,5]. It is very well possible that more light-weight ML approaches would be more suitable as non-symbolic components of LIPS.

The evaluation is somewhat limited and unconvincing.
- It would be informative to see a "time limit vs proof success rate" plot in addition to the table showing the stats for the 90 min. time limit. This limit is rather high, and I suspect that the symbolic approaches, when they solve an inequality, they do this within the first couple of minutes.
- It is possible that the improvement achieved by LIPS compared to CAD-based approaches is due to the set of carefully handcrafted scaling lemmas that fit the test problems. Also, it seems that the size of the filtered set was selected to fit the test problems best, and there was no separate validation set to fit this parameter.
- It would be good to see more ablations: the effect of symbolic filtering; the effect of special tactics (sum of squares, tangent line).


The problems from the test set are publicly available on the internet, therefore GPT-4o could be trained on those. This raises a question about reliability of the evaluation of LIPS, where GPT-4o is used.

Theorem proving in Lean is not sufficiently introduced, which is important given the ML-oriented audience of ICLR.

The presented approach to an extent is inspiring other combinations of symbolic and ML techniques, but LIPS as a system is very specific to the problem of proving competition-style inequalities; it may not translate to more general / practical problems in any simple way.

Minor remarks:
- The description of the X axis in Figure 4b looks to be wrong.
- Figure 3 shows 19 problems, whereas in the text you write about 20 problems.
- The Y axis of the left plot in Figure 3 should start at 0 not 60.
- "AM-GM inequality" -- you should explain the acronym here.

[1] Blaauwbroek et al.: Graph2Tac: Online Representation Learning of Formal Math Concepts. ICML 2024\
[2] Piotrowski et al.: Machine-Learned Premise Selection for Lean. TABLEAUX 2023\
[3] Zhang et al.: Online Machine Learning Techniques for Coq: A Comparison. CICM 2021\
[4] Gauthier et al.: TacticToe: Learning to Reason with HOL4 Tactics. LPAR 2017\
[5] Urban et al.: MaLARea SG1 -- Machine Learner for Automated Reasoning with Semantic Guidance. IJCAR 2008

**Questions:**

1. Could you provide the Lean and SMT formalizations of the test inequalities?
2. Could you provide the implementation of LIPS?
3. What precise parameters were used for SMT solvers to obtain the CAD baseline?
4. How is MCTS implemented? Do you apply symbolic filtering during the search, similarly as in LIPS?
5. Why do you write "Please reason step by step" in the prompts for neural ranking? What are the full resulting responses from GPT-4o?
6. What is meant by "iteration of LIPS" in Figure 3? And what is an "iteration" in the oracle? How do you obtain the oracle proofs?
7. How does search work in LIPS? Is it best-first search? Or breadth-first search?
8. In line 412 you write: "LIPS outperforms the existing method in 13 out of 20 problems". I cannot see this in Figure 3, could you explain?
9. When you do neural rewriting, how do you verify that the rewrite is correct?
10. You use SymPy's `simplify` to compare with "neural" rewrites. What does this function do?
11. Did you consider to use simpler ML techniques in place of GPT-4o?
12. What would happen if you get rid of the neural ranking and use solely homogeneity and decoupling metrics to rank the goals?
13. You write: "Our techniques could serve as a solid foundation for broader areas of mathematical research such as
information theory, optimization, and deep learning". Could you explain a bit how?

---

> ### Author Response · Authors · 2024-11-18
> **Rebuttal by Authors (Part 1)**
>
> **Dear reviewer cEd1:**
>
> Thank you for taking the time to review our paper and provide feedback! Below we address your questions and concerns. Please feel free to follow up if you have further questions!
>
> **1. Data and Code Availability**
>
> We wholeheartedly agree with you on the importance of open source. We will release all the data and code of this paper (pending compliance review), including the test datasets, the implementation of LIPS and MCTs, as well as our Lean and SMT formalizations.
>
> **2. Possibility of Unintended Errors When Formalizing Math Problems**
>
> Our problem domain is immune to many formalization errors plaguing existing benchmarks such as MiniF2F [1]. A common source of errors arises from issues with natural numbers or integers in Lean 4, e.g., [algebra_ineq_nto1onlt2m1on](https://leanprover.zulipchat.com/#narrow/channel/219941-Machine-Learning-for-Theorem-Proving/topic/Aristotle/near/451234781). The inequalities proved by LIPS are not affected by such errors because they only involve real variables, without integers or natural numbers. Furthermore, we have manually checked the formalization of inequalities in our test datasets and plan to release them upon publication.
>
> [1] Zheng, Kunhao, Jesse Michael Han, and Stanislas Polu. "Minif2f: a cross-system benchmark for formal olympiad-level mathematics." arXiv 2021
>
> **3. Algorithm for Proof Search**
>
> We used best-first search, with LLMs for ranking the open proof goals in each iteration (Sec. 4.2). We will revise the paper to describe the proof search algorithm explicitly and in more detail. Thank you for the suggestions!
>
> **4. Missing Related Works**
>
> Thank you for suggesting relevant papers we have missed. We have included them in the revision.
>
> **5. Using Simpler ML Models Instead of LLMs**
>
> In principle, it is possible to replace LLMs in LIPS with simpler models. However, we would have to solve two challenges: First, the model must generate effective rewriting tactics from an infinite space of possible rewrites. Second, it must learn from limited data. Pre-LLM models typically struggle with open-ended generation and zero-shot prediction. Therefore, using them in LIPS may require significant redesign of the method.
>
> **6. Time Limit vs. Proof Success Rate**
>
> For LIPS, We present the proving success rates under different time limits in the table below. The results indicate that the success rate of LIPS improves gradually as more time is allocated. Although symbolic solvers (CAD and MMA) verify the inequalities in under 10 minutes, it is important to note that symbolic solvers are unable to construct human-readable proofs for the problems. Additionally, we plan to further enhance the efficiency of LIPS in future versions, for example, by optimizing the SymPy `pattern_matching` function.
>
> | Method | <10min | <30min | <50min | <70min | <90min |
> | ------ | ------ | ------ | ------ | ------ | ------ |
> | LIPS   | 5%     | 15%    | 35%    | 60%    | 80%    |
>
> **7. Selecting Lemmas and the Filtered Size Hyperparameter**
>
> The lemmas used in LIPS' scaling tactics, detailed in Appendix C (LIPS paragraph), were constructed from [1, 2, 3] and are all standard inequalities (e.g., Cauchy-Schwarz, Chebyshev, and Hölder) commonly used by human contestants in math competitions. None of the lemmas is tied to any particular problem in the test set.
>
> The size of the filtered set is primarily influenced by the LLM's ability to accurately identify a promising proof candidate. In practice, determining the optimal size using a validation set can be computationally expensive. In our experiment, we adopt the [``needle in a haystack’’](https://towardsdatascience.com/the-needle-in-a-haystack-test-a94974c1ad38) approach. Starting with an inequality (e.g., $ab+bc+ca \leq a^2 + b^2 + c^2$) as the initial seed, we manually generated a series of proof candidates. The size of the proof candidates was incrementally increased until the LLM could no longer reliably select the best candidate.
>
> [1] Mildorf, Thomas J. "Olympiad Inequalities." Preprint, https://artofproblemsolving.com/articles/files/MildorfInequalities.pdf
>
> [2] Wei, Chenrui, Mengzhou Sun, and Wei Wang. "Proving Olympiad Algebraic Inequalities without Human Demonstrations." NeurIPS Datasets and Benchmarks Track 2024
>
> [3] Kozma, L. Useful Inequalities. Preprint, https://www.lkozma.net/inequalities_cheat_sheet/ineq.pdf
>
> **8. Potential Data Contamination Due to the Use of GPT-4o**
>
> Please see our common response at the top of this page.
>
>
> **9. Ablations of Symbolic Filtering, Sum-of-squares and Tangent Line**
>
> Please see additional ablations in the common response at the top of this page.
>
> **10. Minor Errors and Writing Suggestions**
>
> Thank you for your detailed comments. We will incorporate them into the next revision, adding more introduction to theorem proving in Lean, fixing Figure 3 and 4, as well as spelling out the full names of acronyms such as AM-GM inequality.
>
>
> *(Connect to Part 2)*

---

> > ### Comment · Reviewer_cEd1 · 2024-11-21
> >
> > > We will release all the data and code of this paper
> >
> > I find it somewhat strange that you cannot release all the data now for inspection.
> > Could you at least release the SMT formalisations now? Or could you release e.g. 10 SMT problems that cvc5 and z3 couldn't solve but LIPS could?
> >
> > > We present the proving success rates under different time limits in the table below.
> >
> > Could you do the same for CAD and MMA, using higher time resolution in the table, or preferably presented in a plot similar to these?
> > https://tptp.org/CASC/J12/WWWFiles/ResultsPlots.html

---

> ### Author Response · Authors · 2024-11-18
> **Rebuttal by Authors (Part 2)**
>
> **Response to Additional Questions**
>
> Thank you for your detailed questions. We address them individually.
>
> > Q1: Could you provide the Lean and SMT formalizations of the test inequalities?
> >
> > Q2: Could you provide the implementation of LIPS?
>
> Yes, we plan to open source the data and code of this paper.
>
>
> > Q3: What precise parameters were used for SMT solvers to obtain the CAD baseline?
>
> For z3, we use default parameters. For cvc5, we use default parameters except that [`nl-cov-force`](https://cvc5.github.io/docs/cvc5-1.2.0/options.html#lbl-option-nl-cov-force) is set to `True` for enabling CAD.
>
>
> > Q4: How is MCTS implemented? Do you apply symbolic filtering during the search, similarly as in LIPS?
>
> Our MCTS implementation follows the approach introduced in AIPS [1], with details described in Appendix C. Symbolic filtering is not used during MCTS.
>
> [1] Wei et al. "Proving Olympiad Algebraic Inequalities without Human Demonstrations." NeurIPS Datasets and Benchmarks Track 2024
>
> > Q5: Why do you write "Please reason step by step" in the prompts for neural ranking? What are the full resulting responses from GPT-4o?
>
> Instructing LLMs to reason step by step was introduced by Kojima et al. [1] and is widely used for eliciting chain-of-thought in LLM’s response without providing explicit chain-of-thought examples in the prompt.
>
> For the input prompt in Sec. 4.2, GPT-4o’s output is:
>
> ```
> To rank the transformations in terms of their ability to reduce the proving difficulty, we need to evaluate each transformation step by step.
>
> ### Original Inequality
> The original inequality is:
> \[
> \frac{1}{a^2+2} + \frac{1}{b^2+2} + \frac{1}{c^2+2} \leq \frac{1}{3a^2+3b^2+c^2} + \frac{1}{3b^2+3c^2+a^2} + \frac{1}{3a^2+3c^2+b^2}
> \]
> with the condition \(a^2 + b^2 + c^2 = 1\).
>
> ### Transformations
>
> #### Transformation (1)
> \[
> \frac{1}{a^2 + 2} + \frac{1}{b^2 + 2} + \frac{1}{c^2 + 2} \leq 2 \sqrt{\frac{1}{a^2 + 3b^2 + 3c^2} \cdot \frac{1}{b^2 + 3a^2 + 3c^2}} + \frac{1}{c^2 + 3a^2 + 3b^2}
> \]
> - This transformation involves using the AM-GM inequality to combine terms, which simplifies the comparison of left and right sides.
> - It leverages symmetry and reduces the complexity slightly by introducing a square root, which can be easier to handle in some contexts.
>
> #### Transformation (2)
> \[
> \frac{1}{a^2 + 2} + \frac{1}{b^2 + 2} + \frac{1}{c^2 + 2} \leq \frac{9}{7a^2 + 7b^2 + 7c^2}
> \]
> - This transformation simplifies the problem by reducing the right-hand side to a single fraction.
> - It takes advantage of the condition \(a^2 + b^2 + c^2 = 1\), simplifying the denominator on the right-hand side to a constant value \(7\).
>
> #### Transformation (3)
> \[
> \frac{1}{a^2 + 2} + \frac{1}{b^2 + 2} + \frac{1}{c^2 + 2} \leq \frac{1}{3 - 2a^2} + \frac{1}{3 - 2b^2} + \frac{1}{3 - 2c^2}
> \]
> - This transformation attempts to reframe the right-hand side using \(3 - 2a^2\), \(3 - 2b^2\), and \(3 - 2c^2\).
> - The symmetry of this transformation is useful, but it does not significantly simplify the structure of the inequality.
>
> ### Ranking
> 1. **Transformation (2)** is the most meaningful because it reduces the entire right side to a constant dependent only on the given condition, which significantly simplifies proving the inequality.
> 2. **Transformation (1)** is also a meaningful transformation, as it uses AM-GM inequality to introduce a symmetric form that could be easier to manage.
> 3. **Transformation (3)** is the least meaningful in terms of simplifying the proving process, as it does not reduce the complexity as effectively as the others.
>
> Thus, the ranking in descending order of transformation effectiveness is:
> \[
> \boxed{\{(2), (1), (3)\}}
> \]
> ```
>
> [1] Kojima, Takeshi, et al. “Large language models are zero-shot reasoners.” NeurIPS 2022
>
> > Q6: What is meant by "iteration of LIPS" in Figure 3? And what is an "iteration" in the oracle? How do you obtain the oracle proofs?
>
> During best-first search, each iteration involves using the tactic generator to expand the most promising proof goal. Consequently, “Iteration of LIPS” refers to the number of times the tactic generator is invoked. “Iteration” in the oracle means the number of tactics in the final proof. Thank you for pointing out the confusion. We will clarify these terms in the next revision.
>
> > Q7: How does search work in LIPS? Is it best-first search? Or breadth-first search?
>
> Best-first search, as addressed in a previous remark.
>
> > Q8: In line 412 you write: "LIPS outperforms the existing method in 13 out of 20 problems". I cannot see this in Figure 3, could you explain?
>
> In Figure 3, the X-axis represents the indices of all problems successfully proved by LIPS, totaling 16 problems. As illustrated in Figure 3(a), LIPS underperforms the existing method on only three problems (indexed as 10, 15, and 16). Thus, we conclude that LIPS outperforms the existing method on at least 13 problems. We will clarify it in the revision.
>
> *(Connect to Part 3)*

---

> ### Author Response · Authors · 2024-11-18
> **Rebuttal by Authors (Part 3)**
>
> > Q9: When you do neural rewriting, how do you verify that the rewrite is correct?
>
> Please see our common response at the top of this page.
>
> > Q10: You use SymPy's simplify to compare with "neural" rewrites. What does this function do?
>
> SymPy’s `simplify` function automatically simplifies a given mathematical expression using heuristics. Details and examples can be found in [its documentation](https://docs.sympy.org/latest/modules/simplify/simplify.html#sympy.simplify.simplify.simplify).
>
> > Q11: Did you consider to use simpler ML techniques in place of GPT-4o?
>
> As discussed in a previous remark, it is difficult to directly replace LLMs in LIPS with simpler ML models, due to challenges in rewriting tactic generation and zero-shot prediction.
>
> > Q12: What would happen if you get rid of the neural ranking and use solely homogeneity and decoupling metrics to rank the goals?
>
> Please see this ablation in our common response at the top of this page.
>
> > Q13: You write: "Our techniques could serve as a solid foundation for broader areas of mathematical research such as information theory, optimization, and deep learning". Could you explain a bit how?
>
> We meant LIPS could potentially be extended to help prove theorems from many branches of advanced mathematics such as information theory and optimization. Please see our common response at the top of this page for more discussion and examples.

---

> ### Comment · Reviewer_cEd1 · 2024-11-21
>
> Now I also think that more fair comparison to CAD/SMT-based methods would involve appending the pre-designed 96 scaling
> tactics and 16 rewriting tactics to the input inequality problems provided to the SMT solvers. Could you comment on that? Perhaps the lower CAD performance is because the SMT solvers did not have the access to the prior knowledge encoded in the pre-designed tactics?
>
> Also, I agree with you that the symbolic methods you test are not outputting interpretable proofs, but it does not mean that they cannot be made interpretable/readable. Perhaps you could also rewrite/reconstruct the SMT proofs into Lean proofs.

---

> > ### Author Response · Authors · 2024-11-24
> >
> > Thank you for following up.
> >
> > > I find it somewhat strange that you cannot release all the data now for inspection.
> >
> > Since we have co-authors from industry labs, the data release is subject to their employer's legal and privacy review process. We do not anticipate any issue during the review and will open source as soon as possible after the review.
> >
> > > could you release e.g. 10 SMT problems that cvc5 and z3 couldn't solve but LIPS could?
> >
> > As suggested, we provide ten problems [here](https://figshare.com/s/fec6a646d814749b07a5), which include SMT formalizations, Lean 4 formalizations, and LIPS solutions.
> >
> > > Could you do the same for CAD and MMA, using higher time resolution in the table
> >
> > Thanks for the comment. In the provided [link](https://figshare.com/s/fec6a646d814749b07a5), we also include CPU time vs. solution count curves for the symbolic solvers CAD and MMA.
> >
> > > appending the pre-designed 96 scaling tactics and 16 rewriting tactics to the input inequality problems provided to the SMT solvers … Perhaps the lower CAD performance is because the SMT solvers did not have the access to the prior knowledge encoded in the pre-designed tactics?
> >
> > We are not sure if we understand what is being suggested, as SMT solvers cannot take tactics as input. To make an educated guess, we think the reviewer might be suggesting adding the lemmas used by scaling tactics (i.e., existing inequalities such as AM-GM and Cauchy Schwarz) to SMT’s input. To investigate this possibility, we conducted additional experiments by incorporating the scaling lemmas into the SMT input as universally quantified formulas (e.g., $\forall x, y, x^2 + y^2 \geq 2xy$). We evaluated the impact of this approach using different SMT and CAD systems, including Z3, CVC5, and Maple's CAD implementation. The results indicate that incorporating the lemmas either had no effect or degraded the performance.
> >
> > These findings are consistent with our intuition. The CAD algorithm itself is not designed to leverage lemmas. The only potential benefit of providing lemmas to SMT lies in their use during the preprocessing step. During preprocessing, in theory, an SMT solver could utilize these lemmas to simplify the target theorem before invoking CAD.  However, we believe this is unlikely to happen in practice, for two reasons: First, SMT solvers are known to struggle with handling quantified formulas [1]. Second, for simple lemmas such as AM-GM, SMT solvers might recognize them as tautologies (i.e., equivalent to `true`) and discard them in subsequent computation.
> >
> > [1] Yeting Ge and Leonardo De Moura. "Complete instantiation for quantified formulas in satisfiability modulo theories." International Conference on Computer Aided Verification (CAV). 2009
> >
> > > symbolic methods you test are not outputting interpretable proofs, but it does not mean that they cannot be made interpretable/readable. Perhaps you could also rewrite/reconstruct the SMT proofs into Lean proofs.
> >
> > We respectfully disagree. First, formal proofs in proof assistants like Lean are not necessarily human-readable. For example, although the proof of the four color theorem has been formalized in Coq [1], to the best of our knowledge, no human-readable proof of the theorem exists [2]. Second, it's not always possible to extract proofs from SMT solvers and translate them into Lean. When an SMT solver uses CAD, extracting a proof would be challenging, if not impossible. It may require formalizing the entire theoretical foundation and implementation of CAD (see [3] for details).
> >
> > [1] [A Formal Proof of the Four Color Theorem in Coq](https://github.com/coq-community/fourcolor)
> > [2] David S Richeson. "[The Colorful Problem That Has Long Frustrated Mathematicians.](https://www.quantamagazine.org/only-computers-can-solve-this-map-coloring-problem-from-the-1800s-20230329/)" Quanta Magazine
> > [3] Barbosa, Haniel. "Challenges in SMT Proof Production and Checking for Arithmetic Reasoning." SC-Square@ ISSAC. 2023

---

> > > ### Comment · Reviewer_cEd1 · 2024-11-25
> > >
> > > Dear authors, thank you for the response and additional results. I think that the time vs n. of solved problems plots should be included in the paper, at least in the appendix. SMT solvers, as I can see, solve problems very quickly, and this should not be hidden as this is important in practice. I acknowledge, however, that there are situations where one can have longer time limit.
> > >
> > > I tried to run z3 and cvc5 on the 10 provided problems, but bot SMT solvers complained that `pow` is undefined. Could you provide complete problems so that I can test the SMT solvers on them?

---

> > > > ### Author Response · Authors · 2024-11-26
> > > >
> > > > Thank you for your response. As suggested, we will include the time vs. number of solved problems plots in the revised version.
> > > >
> > > > Regarding the SMT formalizations, we integrated [PySMT](https://github.com/pysmt/pysmt) as the SMT API in LIPS, which automatically encodes `pow` as `^`. Additionally, we have updated the problems in the provided [link](https://figshare.com/s/fec6a646d814749b07a5) (in `z3_formalizations` and `cvc5_formalizations`), which can be directly run using the `z3` or `cvc5` commands.

---

> > > > > ### Comment · Reviewer_cEd1 · 2024-11-26
> > > > >
> > > > > Thank you! However, when I run cvc5 I see this:
> > > > > ```
> > > > > ./cvc5 cvc5_formalizations/1.smt
> > > > > cvc5_formalizations/1.smt:1.2: No set-logic command was given before this point.
> > > > > cvc5_formalizations/1.smt:1.2: cvc5 will make all theories available.
> > > > > cvc5_formalizations/1.smt:1.2: Consider setting a stricter logic for (likely) better performance.
> > > > > cvc5_formalizations/1.smt:1.2: To suppress this warning in the future use (set-logic ALL).
> > > > > (error "Parse Error: cvc5_formalizations/1.smt:7.22: expecting same arithmetic types to POW")
> > > > > ```
> > > > > Still it seems to be some problem. (I used cvc5 compiled from the [official code](https://github.com/cvc5/cvc5).)

---

> > > > > > ### Author Response · Authors · 2024-11-26
> > > > > >
> > > > > > The `Parse Error` appears to occur when using CVC5 version `cvc5 1.2.1-dev`. This issue can be mitigated by either (1) using the stable version `cvc5 1.2.0` instead, or (2) explicitly specifying the `set-logic` command in the SMT formalizations.
> > > > > >
> > > > > > We have also updated the CVC5 formalizations in the shared [link](https://figshare.com/s/fec6a646d814749b07a5) by adding `(set-logic NRA)` or `(set-logic NRAT)` to each problem.

---

> > > > > > ### Author Response · Authors · 2024-12-01
> > > > > >
> > > > > > Dear reviewer cEd1,
> > > > > >
> > > > > > Thank you for your valuable feedback. As the discussion period is closing shortly, we want to make sure all your major concerns have been addressed. If you have further questions, please let us know. If your concerns have been resolved, we kindly invite you to consider updating your score accordingly. Thanks again!

---

> > > > > > > ### Comment · Reviewer_cEd1 · 2024-12-02
> > > > > > >
> > > > > > > Dear authors, thank you for all your feedback (in particular the SMT problems -- they worked now), now I will reconsider the score. Are you perhaps able to provide the code and data already? It would make my decision easier.
> > > > > > >
> > > > > > > Also, as a minor remark: in Fig. 3 it is easy to miss that some indices on the x axis are not present -- perhaps you could just order the problems by "tactic pruning ratio" for better visibility and re-index the problems according to it?

---

> > > > > > > > ### Author Response · Authors · 2024-12-03
> > > > > > > >
> > > > > > > > Thank you for the response.
> > > > > > > >
> > > > > > > > > Are you perhaps able to provide the code and data already?
> > > > > > > >
> > > > > > > > The code and data are still undergoing a review process, which typically takes a few months to complete. We are committed to open-sourcing both the code and data once the review is finalized. Transparency and accessibility are priorities for us, and we see no reason not to share these resources.
> > > > > > > >
> > > > > > > > > Also, as a minor remark: in Fig. 3 it is easy to miss that some indices on the x axis are not present -- perhaps you could just order the problems by "tactic pruning ratio" for better visibility and re-index the problems according to it?
> > > > > > > >
> > > > > > > > Thanks for the comment. We will update Figure 3 accordingly.

---

> > > > > > > > > ### Comment · Reviewer_cEd1 · 2024-12-03
> > > > > > > > >
> > > > > > > > > Thank you! I've raised my score to 5. I appreciate the engagement of the authors in the discussion. I also think that combining LLMs with formal theorem provers is a very sensible research avenue. However, in my opinion the paper currently has several weaknesses which do not allow me to give more positive score. Here I mainly mean somewhat limited evaluation, not enough ablations, the question about data contamination, the question about generality and impact of the presented approach.

---

> > > > > > > > > > ### Author Response · Authors · 2024-12-03
> > > > > > > > > >
> > > > > > > > > > Thank you, and in particular for your thoughtful involvement in the discussion! Your insightful feedback and valuable suggestions will greatly help us improve our paper. We truly appreciate it!

---

### Official Review · Reviewer_4dY9 · 2024-11-04

**Soundness:** 3
**Presentation:** 3
**Contribution:** 4
**Rating:** 8
**Confidence:** 3

**Summary:**

The paper studies a neural-symbolic integration method, named Lips, for mathematical inequality problems. The paper first identifies the two key tactics in inequality problems, scaling and rewriting. Application of the scaling tactic is determined by a symbolic method, and application of the rewriting tactic is by prompting an LLM. Applying these tactics creates new proof goals, and the paper filters the created goals by a symbolic method and scores the surviving ones by prompting an LLM.

**Strengths:**

+ Solving mathematical inequalities is a challenging, and significant task. It also has application to formal verification.

+ The proposed approach is a nice integration of symbolic and neural methods. The criterion to distinguish between their use in tactic application is clear: reasoning in a finite search space is conducted symbolically and reasoning in an infinite space is neurally. The idea of symbolic filtering and neural scoring is natural, but its significance has been (partly) proven in the experiment.

+ The experiment evaluates the performance of the proposed approach and the potential with respect to the improvement of the underlying LLM. Both are promising.

**Weaknesses:**

- I don't think this is a weakness of the current form of the work, but the paper could enhance the work and broaden the audience if it can claim there are other math domains with two types of tactics such that one is suitable for symbolic reasoning (or the search space of its application is finite) and the other is for neural reasoning (or its search space is infinite), like scaling and rewriting in inequality problems.

- Some parts of the proposed approach are unclear (see the following questions).

Other comments:

- l296: In the definition of HM, the correction of the second occurrence of $d_i$ may be $d_j$.
- I think, after calculating the decoupling score and homogeneity score, only top-k candidates are selected (in the experiment, it seems that only top-10 candidates remain), but it might be nice to clarify it in Section 4 (I noticed it during reading Section 5.1).
- The caption of the right figure in Figure 4 is a bit confusing. Rephrasing it to, e.g., "Various sizes of filtered goal sets" may be helpful.
- For RQ2, the paper says a direct comparison of proving time could be unfair, but I'm not sure whether only comparing the pruning ratio of scaling tactics and the number of iterations in goal selection is fair. Even if the pruning and goal selection are improved over a long period of time, it may not lead to improving the performance of the prover itself. I'm wondering that showing all the comparisons of proving time, pruning ration, and the number of iterations in goal selection may be more fair.

**Questions:**

- How influential on the performance are the symbolic-based transformation tactics, sum-of-squares and tangent line tricks? In other words, what happens if only neural-based transformation is applied?

- What are the up-arrow and down-arrow in the caption of Figure 3?

---

> ### Author Response · Authors · 2024-11-18
> **Rebuttal by Authors**
>
> **Dear reviewer 4dY9:**
>
> Thank you for taking the time to review our paper and provide feedback! Below we address your questions and concerns. Please feel free to follow up if you have further questions!
>
> **1. Comparison with AIPS in Proving Time**
>
> First, we would like to clarify that the code for AIPS is still unavailable; therefore, we have re-implemented its tactic pruning based on the descriptions in the paper. As suggested, we also evaluated proving time by replacing our tactic pruning methods with AIPS’s equality-checking approach. The results below (similar to Figure 3) show that LIPS’s pruning consistently outperforms AIPS’s on 14 out of 20 problems in terms of proving time (minutes). Moreover, AIPS’s pruning significantly degrades LIPS’ proving success rate on MO-INT from 80% to 40%.
>
> | Method         | 1      | 2      | 3      | 4      | 5      | 6    | 7    | 8      | 9      | 10     | 11     | 12   | 13     | 14     | 15     | 16    | 17     | 18     | 19     | 20   |
> | -------------- | ------ | ------ | ------ | ------ | ------ | ---- | ---- | ------ | ------ | ------ | ------ | ---- | ------ | ------ | ------ | ----- | ------ | ------ | ------ | ---- |
> | AIPS's pruning | 60     | 90     | 90     | 90     | 90     | 90   | 90   | 68     | 90     | **49** | 90     | 90   | 40     | 87     | 33     | **2** | 90     | 90     | 77     | 90   |
> | LIPS's pruning | **43** | **58** | **77** | **74** | **44** | 90   | 90   | **15** | **79** | 69     | **82** | 90   | **31** | **68** | **19** | 6     | **58** | **53** | **34** | 90   |
>
>
> **2. Neuro-Symbolic Reasoning for Other Mathematical Domains**
>
> In LIPS, we design two types of tactics: scaling tactics are more natural for symbolic reasoning, and rewriting tactics are more natural for neural networks. This kind of structure is not limited to the domain of inequalities. One example is AlphaGeometry for proving 2D Euclidean geometry problems [1]. It uses symbolic reasoning to deduce basic geometric properties and neural networks to introduce auxiliary constructions (points, lines, or circles not present in the original problem). For a broader discussion on the generality of LIPS, please see our common response at the top of this page.
>
> [1] Trinh, Trieu H., et al. "Solving olympiad geometry without human demonstrations." Nature 2024
>
> **3. Ablations of Sum-of-squares and Tangent Line Tricks**
>
> Please see additional ablations in common response at the top of this page.
>
>
> **4. Arrows in Figure 3 Caption**
>
> The up arrow ($\uparrow$) means “higher is better” (for the tactic pruning ratio). The down arrow ($\downarrow$) means “lower is better” (for the number of iterations). Thank you for pointing out the confusion. We will clarify them in the next revision.
>
>
> **5. Typos and Writing Suggestions**
>
> Thank you for your valuable feedback! We have revised accordingly.

---

> > ### Comment · Reviewer_4dY9 · 2024-11-20
> > **Response**
> >
> > Thank you for the response. It clarifies the concerns and questions in my review. Although we need to be careful of the potential issue with AIPS being reimplemented, the comparison with AIPS seems impressive in that some problems which the reimplemented AIPS cannot solve are solved by LIPS.

---

### Author Response · Authors · 2024-11-18
**Overall Response (Part 1)**

**Dear reviewers and meta-reviewers:**

Thank you for your time and constructive feedback on our work. We're encouraged that reviewers find LIPS a well-integrated neuro-symbolic approach that is highly effective at solving challenging inequality problems in math competitions. Below we first address questions raised by multiple reviewers, followed by detailed responses to each reviewer individually. We look forward to follow-up discussions!

*(Connect to Part 2)*

---

> ### Author Response · Authors · 2024-11-18
> **Overall Response (Part 2)**
>
> **1. Broad Impact and Generality of LIPS (4dY9, cEd1, v6PZ, YrrG)**
>
> While LIPS focuses on a concrete domain—inequality problems in pre-college math competitions (described in Figure 1)—it makes important contributions and can potentially generalize to other domains. First, the inequality problems tackled by LIPS are important beyond competition-level mathematics. They are closely related to quantifier-free non-linear real arithmetic ([QF_NRA](https://smt-lib.org/logics-all.shtml#QF_NRA)) in SMT (Satisfiability Modulo Theories), which plays a foundational role in formal verification [1]. One potential application of LIPS is to enhance SMT solvers by serving as a QF_NRA backend.
>
> Furthermore, inequalities are pervasive not only in pre-college math competitions but also in many branches of advanced mathematics, including analysis, statistics, and information theory. For example, a recent breakthrough in partial differential equations (PDEs)—the proof of blowup for the Euler equations [2]—relied heavily on inequalities. As highlighted by a Quanta Magazine article [3], "Hou and Chen had to prove the statement, which boiled down to a complicated inequality involving terms from both the re-scaled equations and the approximate solution." Although the inequalities encountered in advanced mathematics often extend beyond the current capabilities of LIPS (e.g., involving calculus), they share many foundational techniques, such as rewriting and scaling, which are utilized by LIPS. Extending LIPS to handle inequalities in other mathematical domains is a promising direction for future work. To demonstrate the feasibility, we use LIPS to prove [the negativity of the KL divergence between two binomial distributions](https://statproofbook.github.io/P/bin-kl.html)—an important result in information theory. LIPS is able to generate the following proof, which cancels out a nonnegative factor $n$, rearrange the inequality, and apply the Jensen inequality.
>
> ```
> theorem KL_nonnegativity {n a b : ℝ} (ha : a > 0) (ha' : 1 - a > 0) (hb : b > 0) (hb' : 1 - b > 0) (hn : n > 0) : 0 ≤ n * a * log (a / b) + n * (1 - a) * log ((1 - a) / (1 - b)) := by
>  llm_cancel_factor 0 - (n * a * log (a / b) + n * (1 - a) * log ((1 - a) / (1 - b))) = (-n)*(a*log (a/b) + (1-a)*log ((1-a)/(1-b)))
>  llm_simplify 0 - (a * log (a / b) + (1 - a) * log ((1 - a) / (1 - b))) = a * log (b/a) + (1 - a) * log ((1-b)/(1-a))
>  scale NEQ_weighted_Jensen_logx_left (i1 := a) (i2 := (1-a)) (u := (b/a)) (v := ((1-b)/(1-a))) (right := 0)
>  llm_simplify (a + (1 - a)) * log ((a * (b / a) + (1 - a) * ((1 - b) / (1 - a))) / (a + (1 - a))) = 0
> ```
>
> Abstracting away from specific system designs, the principles underlying LIPS are highly general and transferable to other domains. A primary challenge in mathematical theorem proving, unlike board games such as Go, is the infinite action space [4], which resists effective exploration. A key insight of LIPS is that domain knowledge can enable the design of an action space that allows symbolic reasoning to efficiently explore at least a subset of the space. This general principle has been successfully applied in other domains. For instance, AlphaGeometry [5] partitions its action space into two categories: deducing properties from the premises of an existing geometry problem and introducing auxiliary constructions (e.g., points, lines, or circles). The former category is particularly amenable to symbolic reasoning, analogous to the scaling tactics in LIPS.
>
> The reviewers' question on generality is insightful and related to a central debate in AI for Mathematics: *Should we strive to build generalist AI systems that are agnostic to mathematical domains, or should we develop specialist systems tailored to specific domains?* This dichotomy is reflected in recent works—generalist approaches such as LeanDojo [6] and specialist approaches such as AlphaGeometry, FunSearch [7], and LIPS. Mathematics has an infinite action space. To effectively explore it, we must leverage its structure. The generalist approach assumes that AI can learn and exploit domain-specific structures and idiosyncrasies in a general way. While this approach is desirable in the long term, it faces challenges with the current generation of technology. In contrast, specialists rely on human-designed structures informed by mathematical knowledge, e.g., scaling and rewriting tactics in LIPS. This approach is more practical and effective in the short to medium term. Therefore, we believe AI math specialists like LIPS have value and will coexist with generalists for the foreseeable future.
>
> *(Connect to Part 3)*

---

> ### Author Response · Authors · 2024-11-18
> **Overall Response (Part 3)**
>
> **2. Additional Ablations (4dY9, cEd1)**
>
> Thank you for suggesting these ablations. We will incorporate the results into the next revision.
>
> * **Scaling Tactics**: Figure 4 (a) has shown how the performance on the ChenNEQ dataset (Y axis) changes as we increase the number of lemmas used by scaling tactics (X axis). When X = 0 (without scaling tactics), only 3 out of 41 theorems can be proved, whereas LIPS can prove 39 theorems.
> * **Symbolic Filtering**: Figure 4 (b) has shown how the performance on the ChenNEQ dataset (Y axis) changes as we increase the size of the filtered set (X). A smaller X means more aggressive symbolic filtering. When X is large, e.g., 32, the performance drops under 40%, highlighting the importance of symbolic filtering. We did not test values of X beyond 32, as this consistently caused GPT-4o to hit the maximum token limit (set at 4096)
> * **Sum-of-squares and Tangent Line Tricks**: We newly evaluated LIPS w/o sum-of-squares and tangent lines on ChenNEQ. Among the 39 out of 41 proved inequalities, only 1 requires sum-of-squares, and 3 requires the tangent line trick.
> * **Neural Ranking**: Figure 5 presents an ablation analysis where neural ranking is replaced with random selection. The results demonstrate a significant drop in LIPS performance, from 95.1% to 41.2%, underscoring the necessity of neural ranking. Additionally, we plan to incorporate a new ablation analysis that examines the effect of replacing neural ranking with purely symbolic filtering.
>
> **3. Potential Data Contamination Due to GPT-4o (cEd1, v6PZ)**
>
> We agree with the reviewers on the importance of vigilance against data contamination when evaluating LLM-based methods. However, our analysis indicates that data contamination is unlikely to be a significant contributor to LIPS’s performance.
>
> In Sec. 5.2 (RQ1), we asked OpenAI o1 to prove the 20 problems from the MO-INT dataset and manually evaluated the generated proofs. o1 wasn’t able to prove any one of them. We repeated this experiment using GPT-4o, and the results were the same, indicating that LLMs did not memorize the proofs of test problems during pretraining. In addition, even if they had memorized the (human-written) proofs, it is difficult to translate them from English/LaTex into a formal language like Lean, i.e., autoformalization.
>
>
>
> **4. Implementation Details of Rewriting Tactics in Lean 4 (cEd1, v6PZ)**
>
> We implemented LLM-generated rewriting using 16 Lean 4 tactics. For the complete list, please refer to Appendix C (LIPS paragraph). We will release all the data and code of this paper (pending compliance review), including the Lean 4 implementation of these tactics.
>
> The neural rewriting essentially attempts to transform the current inequality into an equivalent form, which is generated by LLMs. The rewriting tactic in Lean 4 then only needs to verify whether this new form is provable. For instance, given the original inequality $ab+bc+ca \leq a^2+b^2+c^2$, the LLM might produce a new version like $3ab+3bc+3ca \leq (a+b+c)^2$. The rewriting tactic is then invoked to check whether the transformation $(ab+bc+ca) - (a^2+b^2+c^2) = (3ab+3bc+3ca) - (a+b+c)^2$ holds.
>
> To achieve this, we integrate a series of Lean 4 automation tactics under the `llm_simplify` function:
>
> ```lean
> macro "llm_simplify " h:term : tactic =>
>     `(tactic| (apply move_left;
>                 rwp [show $h by
>                 (first
>                 | ·field_simp [*]
>                 | ·(field_simp; ring)
>                 | ·(field_simp [*]; ring)
>                 (try simp only [tsub_le_iff_right]);
>                 (try apply tsub_le_iff_right_p);
>                 norm
>               ))
> ```
>
> Other rewriting tactics follow a similar structure.
>
>
> [1] De Moura, Leonardo, and Nikolaj Bjørner. "Satisfiability modulo theories: introduction and applications." CACM 2011
>
> [2] Chen, Jiajie, and Thomas Y. Hou. "Stable nearly self-similar blowup of the 2D Boussinesq and 3D Euler equations with smooth data I: Analysis." arXiv 2022
>
> [3] Jordana Cepelewicz. "Computer Proof ‘Blows Up’ Centuries-Old Fluid Equations." Quanta Magazine 2022
>
> [4] Poesia, Gabriel, and Noah D. Goodman. "Peano: learning formal mathematical reasoning." Transactions of the Royal Society A 2023
>
> [5] Trinh, Trieu H., et al. "Solving olympiad geometry without human demonstrations." Nature 2024
>
> [6] Yang, Kaiyu, et al. "Leandojo: Theorem proving with retrieval-augmented language models." NeurIPS 2024
>
> [7] Romera-Paredes, Bernardino, et al. "Mathematical discoveries from program search with large language models." Nature 2024

---

> > ### Comment · Reviewer_cEd1 · 2024-11-21
> >
> > Dear authors, thank you for the response. I have some further questions:
> >
> > You write
> >
> > > Figure 5 presents an ablation analysis where neural ranking is replaced with random selection.
> >
> > As far as I understand, Figure 5 shows situation when you remove *both neural ranking and symbolic filtering*. Is that true? In that case it is misleading. I find it crucial to show ablation where you remove neural ranking only and use the symbolic filtering method to provide the ranking. I simply suspect that the neural rewriting may not add any performance here, and the symbolic filtering is enough.
> >
> > > our analysis indicates that data contamination is unlikely to be a significant contributor to LIPS’s performance.
> > > [...] LLMs did not memorize the proofs of test problems during pretraining.
> >
> > I remain sceptical here. Even though LLM's could not provide a full proof, they did see the the proofs, and could memorize certain steps, which may contribute to the performance of LIPS.

---

> > > ### Author Response · Authors · 2024-11-24
> > >
> > > Thank you for the comment.
> > >
> > > > Figure 5 shows situation when you remove both neural ranking and symbolic filtering. Is that true?
> > >
> > > No, the ablation in Figure 5 only replaced neural ranking by random selection, with symbolic filtering applied to both cases.
> > >
> > >
> > > > Even though LLM's could not provide a full proof, they did see the the proofs, and could memorize certain steps, which may contribute to the performance of LIPS.
> > >
> > > We appreciate the reviewer's thoughtful concern regarding data contamination. While our analysis cannot completely rule out this possibility, we view it as a sanity check—capable of identifying potential contamination but not proving its absence. This limitation applies to any method relying on closed LLMs like GPT. If the reviewer's concern pertains broadly to methods using GPT, we agree this is an important issue, albeit one outside the scope of this paper. However, if the concern of data contamination is specific to LIPS, we would be grateful for your further clarification. Additionally, we welcome suggestions for more analyses that might help address your concerns.
> > >
> > > As another check, we manually reviewed selected proofs LIPS generated and observed that many (e.g., the first two proofs in Appendix E) employed different approaches from the corresponding human-written informal proofs.

---

### Meta-Review · Area_Chair_9pFL · 2024-12-20

**Metareview:**

This paper aims to solve Olympiad inequalities using a tool combining symbolic solvers and LLMS, in a neuro-symbolic approach. The approach shows good results in comparison to baselines.

Several concerns and weaknesses could be mitigated during the discussion phase. Most of the reviewers agree that this is a promising and timely approach, drawing power from the neurosymoblic reasoning for relevant problems.

**Additional Comments On Reviewer Discussion:**

The authors and several reviewers engaged in the discussion and could resolve most of the issues.

---

### Decision · Program_Chairs · 2025-01-22

Accept (Poster)